# OpenPhase: Condition-Aware Exploration of Multicomponent Biosystem Phase-Separating Behavior

## Abstract

Liquid-liquid phase separation (LLPS) is a fundamental biophysical process in which biomolecules, such as proteins, DNA and RNA, demix from solution to form distinct liquid phases. Crucially, experimental conditions, such as salt, temperature, pH and concentration, profoundly influence LLPS properties and dynamics, often determining whether phase separation occurs and modulating the propensity of the resulting biomolecular condensates. While numerous machine learning methods have been developed for protein phase behavior prediction, their capabilities are frequently constrained by the inherent complexity of biosystems and the vast variability of environmental conditions. Here, we introduce **OpenPhase**, the first condition-aware platform for system-level exploration of biomolecular phase behavior. OpenPhase uniquely provides well-structured and ready-to-use datasets of experimentally verified phase outcomes coupled with corresponding conditions. We also formalize three canonical tasks (1) condition-aware phase outcome prediction, (2) condition inference of LLPS, and (3) conditional phase-separating system design. These tasks well articulate the interplay between system components, environmental conditions, and emergent phase properties. For each task, we propose novel solutions and benchmark them against strong baseline models. OpenPhase also includes a user-friendly API to facilitate *in-silico* development and evaluation of novel machine learning methods for complex biosystem phase behavior modeling.

## 1 Introduction

Liquid-liquid phase separation (LLPS) is a widespread mechanism by which cells organize their internal environment, leading to the formation of membraneless organelles and biomolecular condensates Banani et al. (2017); Alberti et al. (2019); Xu et al. (2024b). While early studies focused on protein-driven phase separation, it is now clear that many condensates are multicomponent systems, including proteins, DNA, and RNA Laflamme & Mekhail (2020); Gu et al. (2022); Welles et al. (2024). The interplay among these biomolecules, together with their sequence patterns and physicochemical properties, gives rise to complex phase behaviors that are essential for diverse cellular functions and are implicated in diseases Ding et al. (2024). The complexity of these interactions has motivated the development of computational methods to predict phase-separating proteins.

The ability of a system to undergo phase separation is believed to be encoded in the components themselves while extrinsic conditions can modulate the phase outcome to a large extent Singh (2024). Early approaches relied on sequence heuristics as well as recent advances in machine learning or deep learning-based sophisticated models capture subtle patterns in protein sequences, subgroups and structures associated with phase separation to model the phase separation. Despite such advances, predicting phase behavior in multicomponent systems under diverse experimental conditions remains a significant challenge for computational methods. Most methods are trained on binary protein phase separation labels and do not generalize to systems involving nucleic acids or to condition-dependent behaviors Hou et al. (2024); Chu et al. (2022); Mészáros et al. (2020); Feng et al. (2024a). Another concern arises because almost every computational predictor only consider protein sequence or structure information and ignore the effects of experimental conditions Hou et al. (2024); Feng et al. (2024b). Studies have revealed that phase behaviors are highly sensitive to

experimental conditions, such as salt concentration, temperature, pH, and the presence of crowding agents Wang et al. (2022); Arter et al. (2022); Alberti et al. (2018). These factors can modulate the phase behavior, leading to cooperative or competitive phase separation, selective partitioning, and the emergence of multiphase condensates Laflamme & Mekhail (2020). A possible reason for this limitation is that existing datasets and tools often lack entries with quantifiable experimental conditions, making it difficult to train or evaluate models that account for condition-dependent phase behavior. Among existing tools, Droppler is currently the only method that explicitly incorporates experimental conditions into phase separation prediction Raimondi et al. (2021).

To address these issues, we introduce **OpenPhase**, an open-source dataset and package for fast knowledge discovery of phase-separating behavior in multicomponent biosystems. OpenPhase formulates it as three key computational tasks: (1) phase outcome prediction under specific experimental conditions, (2) phase diagram simulation for given multicomponent systems, and (3) phase-separating system protein (system) design under user-defined conditions. Our dataset covers wide range of proteins, nucleic acids, and experimental conditions from three independent sources, enabling benchmarking and development of computational tools. OpenPhase provides a user-friendly interface and supports integration with existing machine learning frameworks. By reformulating *in-silico* prediction as the interplay among system components, phase outcomes, and experimental conditions, OpenPhase provides the foundation for more generalizable pattern of realistic biomolecular phase separation modeling.

**Main contributions:** (i) We introduce OpenPhase, the first systematic framework for modeling biomolecular phase behavior by three key computational tasks: condition-aware phase outcome prediction, condition inference for phase systems, and phase system design, providing a comprehensive approach to understanding phase (ii) We present three well-curated for diverse multicomponent systems with detailed experimental conditions to accelerate machine learning-based discovery of biomolecular phase study. (iii) We present a novel attention-based layer to captures the interactions between biomolecular components and their environmental conditions and achieve the state-of-the-art performance. (iv) For the first time, We simulate the phase diagrams for a given biomolecular systems in a rational approach. (v) We demonstrate the experimental condition-aware generation for phase-separating proteins using diffusion-based models.

## 2 PROBLEM FORMULATION

We formulate the exploration of phase-separating biosystems within a unified context so that we can model the interplay among three key variables:

- **System Components ($x$):** A set of representations of the components. Common representations include but are not limited to sequence one-hot vector, intrinsic biophysical characteristics and language model embeddings derived from pretrained models.

- **Experimental Conditions ($c$):** A set of representations of extrinsic environmental factors like salt concentration, temperature, solute concentration, crowding agent and pH.

- **Phase Outcome ($y$):** A binary variable $y \in \{0, 1\}$ (or a quantity $y \in [0, 1]$ from continuous measure from phase experiments) indicating whether (or how intense) the system undergoes phase separation (dual-phase, $y = 1$) or remains in a single phase ($y = 0$) under given experiment conditions.

We also formulate three canonical tasks that are crucial for understanding phase separation of a biomolecular system. These tasks explain basic aspects of the interplay between system components, experimental conditions, and phase outcomes in the biocondensate study. The tasks are formulated as follows:

- **Task 1 - Condition-aware phase outcome prediction:** Given a biological system representation ($x$) and a set of experimental conditions ($c$), the objective is to predict the phase outcome $y$. This predictive capability $p(y|x, c)$ is essential for simulating phase diagrams and assessing how a system will behave under untested conditions.

- **Task 2 - Condition inference for phase system:** With system representations ($x$) and phase outcome ($y$), the goal is to infer the potential experimental conditions ($c$) that could

have produced this result. This corresponds to computing the posterior probability distribution $p(c|x, y)$. This task is crucial for hypothesis generation, enabling exploration of key environmental factors that drive phase separation.

- **Task 3 - Phase system design:** Given a desired phase outcome ($y$) and a target set of experimental conditions ($c$), the task is to generate potential system components ($x$), primarily proteins, that are likely to exhibit the observed behavior. This is a conditional generative modeling problem that requires sampling from the distribution $p(x|y, c)$. This task enables the *in-silico* design of novel proteins with adjustable phase sensitivity.

## 3 OPENPHASE DATASET

The OpenPhase dataset is exhibited in a comprehensive, curated, and processed collection of validated phase-separating with variable experimental conditions. It serves as a unified, standardized, and comparable benchmarking resource for biocondensate discovery. Three datasets cover a broad spectrum of diverse systems. Rich annotations and co-existence records of components and conditions enable further systematic benchmarking or development of condition-aware computational tools. By integrating diverse experimental conditions and a wide range of biomolecular components, it accelerates the study of phase-separating behavior.

There are three major revised parts in the dataset: (1) the filtered and numerical *LLPSDBv2* dataset Wang et al. (2022), which contains *in-vitro* LLPS records for proteins, RNA, DNA, and corresponding experimental conditions, named as *db1*; (2) a manually collected LLPS records from laboratory notebooks, named as *db2*. To our knowledge, it is the first resource of its kind, encompassing a wide range of multi-component systems (protein, RNA, or DNA), along with detailed experimental conditions (salt concentration, temperature, pH and etc.). (3) the preprocessed dataset derived from *RNAPSEC* Chin et al. (2024), which focuses on the phase behavior of protein + RNA systems, named as *db3*.

### 3.1 PHASE SYSTEM AND COMPONENTS

Table 1: General component statistics in *db1*, *db2* and *db3*

| Dataset | # Entry | # Protein | # DNA | # RNA |
|---------|---------|-----------|-------|-------|
| *db1*   | 6,005   | 546       | 102   | 279   |
| *db2*   | 1,188   | 14        | 0     | 3     |
| *db3*   | 1,514   | 37        | 0     | 147   |

**The *db1* dataset** is sourced from the LLPSDB v2.0 dataset Wang et al. (2022) dated January 13, 2022. It contains 2,917 entries of phase-separating systems with 586 unique proteins and 6,678 experimental conditions. Compared to the first version of LLPSDB Li et al. (2020), which Droppler Raimondi et al. (2021) was based on, this version has almost tripled the number of entries and doubled the number of conditions. We filtered and combined all studied systems including both "Ambiguous system" and "Unambiguous systems" and classified them into 3 protein-only systems, and 4 protein-nucleic acid systems. For example, protein(1) is a typical system containing sole protein component while protein(2) + RNA is another one with 2 proteins and 1 RNA molecules. The details and statistics for *db1* systems are listed in the Appendix 1.1.

**The *db2* dataset** is manually collected independent data source for phase diagram exploration. We assembled the dataset of 1,188 experimentally validated entries, spanning from *in vitro* to *in vivo* systems, natural proteins to synthetic constructs. There are three subsets in *db2*: *LSD1* Xu et al. (2024a), *synthetic IDRs* and *LAF1 RGG*. Quantitative outcome $y$ (*intensity fraction*) of phase experiments in *synthetic IDRs* is also provided. It is the first condition-dependent phase continuous quantification data with high resolution.

**The *db3* dataset** is a well-processed dataset with numeric condition records for condition-dependent behaviors study of diverse protein-RNA combinations. The experimental information and phase component data were extracted from individual experiment in RNAPhaSep Zhu et al. (2022). We summarize the number of components here in Table 1.

## 3.2 Component embeddings

A variety of embedding methods are included in OpenPhase to represent different components, i.e. proteins, RNA, and DNA (Table 2).

Table 2: Protein and nucleic acid embedding methods

| Embedding method | $d_{\text{protein}}$ | $d_{\text{DNA}}$ or $d_{\text{RNA}}$ |
|---|---|---|
| ESM-C Team et al. (2024)[1] | 1,152 | ✗ |
| MTDP Shang et al. (2024)[2] | 1,280 | ✗ |
| RNAPSEC Chin et al. (2024)[3] | 29 | 97 |
| Dictonary | ✗ | 28 |
| One-hot[4] | $20 \times |L|$ | $4 \times |L|$ |

[1]Pretrained 600M model; [2]Multi-Teacher Distillation model based on UniProtKB Bateman et al. (2023); [3]Sequence-based biophysical features; [4]With sequence length $|L|$.

For proteins, the embeddings ($x_{\text{protein}} \in \mathcal{R}^{d_{\text{protein}}}$) can be derived from pretrained protein language models e.g., *ESM-Cambrian* Team et al. (2024), *MTDP* Shang et al. (2024), biophysical characteristics from protein analysis Bonidia et al. (2022); Chin et al. (2024), or from simpler representations such as one-hot vector over the 20 standard amino acids (useful in protein design).

For nucleic acids, the embeddings $x_{\text{DNA}} \in \mathcal{R}^{d_{\text{DNA}}}$ and $x_{\text{RNA}} \in \mathcal{R}^{d_{\text{RNA}}}$ can be obtained similarly. Trainable dictionary method (e.g. `nn.Embedding` layer in PyTorch Paszke (2019)) are used to map nucleic acid description to dense embeddings. If the DNA or RNA sequence is available, features such as nucleotide composition and sequence periodicity can be extracted Bonidia et al. (2022); Chin et al. (2024).

## 3.3 Experimental condition embeddings

For experimental conditions, we extracted and parsed the conditions aligned with the procedure in Droppler Raimondi et al. (2021) and RNAPSEC Chin et al. (2024). We list the available experimental conditions in Table 3.

Table 3: Available experimental condition embeddings

| Exp. conditions | Droppler[1] | RNAPSEC [2] |
|---|---|---|
| $c_{\text{temperature}}$ | $\{0, 1\}^{10}$ | Category 1-3 |
| $c_{\text{[protein/DNA/RNA]}}$ ($\mu$M) | $\mathcal{R}^+$ | Category 1-5 |
| $c_{\text{[iron]}}$ ($m$M) | $\mathcal{R}^+$ | Category 1-5 |
| $c_{\text{pH}}$ | $(0, 14)$ | Category 1-3 |
| $c_{\text{crowding}}$ | $\{0, 1\}$ | ✗ |

[1]A few modifications made on original Droppler method; [2]Based on 5 quintile values for component conc. and iron strength.

Following the Droppler protocol, temperature, component concentration, ionic strength, pH, and crowding agent are standardized and unified. For RNAPSEC settings, protein/RNA concentration, temperature, ionic strength, and pH are normalized and harmonized. Optional categorical encoding can be applied using either fixed ranges or quantile-based binning as required. Full details of the experimental condition embedding procedure are provided in Appendix 1.2.

## 3.4 Dataset split

We split the each dataset into training and test sets by a few types of methods within a given system (e.g. protein(1)) or given subset. While random splitting is a common baseline, it may not adequately assess a model's ability to generalize to underrepresented or unseen scenarios, such as rare phase outcomes or specific experimental conditions. Therefore, alternative split methods are necessary to evaluate model robustness, mitigate biases from data imbalance, and simulate real-world applications where certain conditions or outcomes may be scarce or entirely absent from the training data.

We propose to split the dataset by **phase outcome** or **experimental conditions**. The split strategies include stratified split, zero-shot split, and few-shot split. Details of the split methods are listed in the Appendix 1.3.

### 3.5 USER-FRIENDLY DATASET INTERFACE

OpenPhase provides a user-friendly interface for accessing and manipulating the dataset by inhering all API from `torch.utils.data.Dataset` Paszke (2019). For fast splicing, viewing and editing operation of the dataset, we provided a `anndata.Anndata` Virshup et al. (2024) interface that allows users to easily access the annotations, embeddings, and other unstructured data for system/subset of interest. Users can easily download the dataset, access individual entries, check corresponding $x$, $c$, and $y$ from different settings. We also provided precomputed embeddings for proteins, RNA, DNA and conditions and predefined dataset split approaches. All the processing and embedding transparency for users with flexibility and extensibility, enabling adaptation for specific needs.

## 4 METHODOLOGY

**Task 1 - condition-aware phase outcome prediction:** we introduce a *Condformer* that integrates component self-attention blocks, component-condition cross-attention blocks Vaswani et al. (2017) and a classification or regression head as shown in Figure 1 (a).

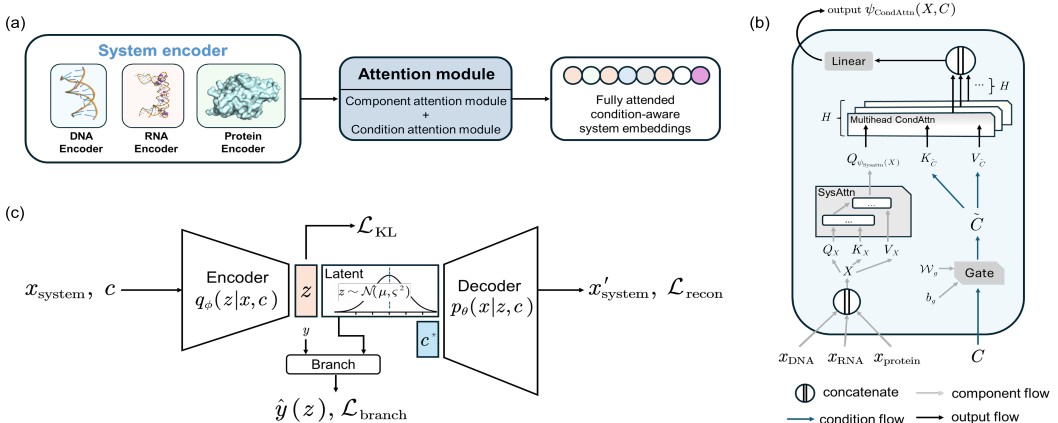

Figure 1: Overview of the proposed methods for the task1 and task2 in OpenPhase. (a) System components $x$ and experimental conditions $c$ feed into the attention-based Condformer; (b) Attention module modeling; (c) Branched conditional variational autoencoder (cVAE) with loss items. $c^*$ is query condition in the inference stage during which all weights are frozen.

The core of our Condformer is the attention mechanism Vaswani et al. (2017); Raimondi et al. (2021), which enables the model to capture dependencies within and between system components and experimental conditions. We employ **self-attention** for every system components and then apply **multi-head cross-attention** for the attended system representation and **gated** condition embeddings. The used scaled dot-product attention and multi-head attention are as below:

$$\text{Attn}(Q, K, V) = \text{softmax}\left(\frac{QK^\top}{\sqrt{d_k}}\right)V \tag{1}$$

$$\text{MultiHead}(Q, K, V) = (h_1 || h_2 || \cdots || h_H)W^O \tag{2}$$

Where $Q \in \mathcal{R}^{d_* \times d_k}, K \in \mathcal{R}^{d_* \times d_k}, V \in \mathcal{R}^{d_* \times d_v}$ are trainable the query, key, and value matrices, respectively, $d_k$ is the dimension of the key vectors, $d_v$ is the dimension of the value vectors. And in multi-head attention, $H$ is the number of attention heads as it in Eq. 1. The concatenated output of all heads is learnable projected through a linear layer $W^O \in \mathcal{R}^{H \times d_k \times d_*}$.

Then given a set of given system-level representations $X \in \mathcal{R}^{n \times d^*}$, the component-attended representation is computed as:

$$\psi_{\text{SysAttn}}(X) = \text{Attn}(Q_X, K_X, V_X) \tag{3}$$

where $X$ is the concatenated embeddings of all system components (e.g. its dimension $d_* = 2d_{\text{protein}} + d_{\text{RNA}}$ for a "protein(2) + RNA" system), and $Q_X = XW^Q$, $K_Q = XW^K$, $V_X = XW^V$ as learnable projection matrices.

A **gated mechanism** Hochreiter & Schmidhuber (1997) that adaptively modulates the influence of unique conditions prior the attention process is introduced. It computes a gating vector $g \in [0, 1]^{d_c}$ for the condition embedding $c$ for our given condition matrix $C \in \mathbb{R}^{n \times d_c}$,

$$g = \sigma(W_g C + b_g)$$
$$\tilde{C} = g \odot C \tag{4}$$

Where $W_g \in \mathbb{R}^{d_c \times d_c}$ and $b_g \in \mathbb{R}^{d_c}$ are learnable parameters, $\sigma(\cdot)$ denotes the sigmoid activation function, and $\odot$ is the element-wise (Hadamard) product. The gated condition embedding $\tilde{C}$ is then used as input to the cross-attention module, allowing the model to selectively emphasize or suppress specific experimental conditions while learning the system context. Then we use multi-head cross-attention to quantify the influence of selected experimental conditions on system components (Figure 1 (b)).

$$\psi_{\text{CondAttn}}(X, C) = \text{MultiHead}(Q_{\phi_{\text{SysAttn}}(X)}, K_{\tilde{C}}, V_{\tilde{C}}) \tag{5}$$

The output of $\psi_{\text{CondAttn}}(X, C)$ is a finial condition-aware representation of the system. For the downstream tasks, we apply either a 2-layer classification or a regression head to infer the phase outcome $y$. Such design enables the model to jointly reason over system components and experimental conditions for variable forms of phase outcome.

**Task 2 - condition inference:** We propose to utilize the conditional variational autoencoder (cVAE) framework with a non-trivial modification of branch structure De Donno et al. (2023); Connor et al. (2021); Salah & Yevick (2024) for condition inference.

Such cVAE models consist of a parameterized encoder network $p_\theta(z|x, c)$ that maps the input $x$ and $c$ into a latent space of latent variables $z$, and a decoder network $q_\phi(z|x, c)$ that reconstructs the input $x$ from the latent variables $z$. We novelly use $y$ to navigate the latent space and gather experiments with the same phase outcome together (Eq. 8). A traditional cVAE is trained to maximize the log-likelihood of the observations given condition $c$, which can be expressed as:

$$p_\theta(x|c) = \int_z p_\theta(x|z, c)p(z|c)\mathrm{d}z \tag{6}$$

where $\theta$ is the model parameter. A cVAE uses an approach that estimates the posterior distribution $q_\phi(z|x, y, c)$, which is a variational approximation to the true posterior $p_\theta(z|x, c)$. And the typical loss function is to jointly minimize the reconstruction error $\mathcal{L}_{\text{recon}}$ and the distributional distance $\mathcal{L}_{\text{KL}}$ between $q_\phi(z|x, c)$ and $p_\theta(z|x, c)$. This is also known as the evidence lower bound (ELBO) objective,

$$\mathcal{L}_{\text{cVAE}}(\theta, \phi) = - \underbrace{\mathbb{E}_{q_\phi(z|x,c)}[\log p_\theta(x|z, c)]}_{\mathcal{L}_{\text{recon}}} + \underbrace{\mathrm{D}_{\text{KL}}(q_\phi(z|x, c)||p_\theta(z))}_{\mathcal{L}_{\text{KL}}} \tag{7}$$

Where $\mathrm{D}_{\text{KL}}$ is the Kullback-Leibler divergence, $\theta$ and $\phi$ are the parameters of the decoder and encoder networks respectively. The branched structure (Figure 1 (c)) is used to optimize the model in a supervised manner, in which a classification head takes latent variable $z$ as input and predicts corresponding phase outcome $y$. The branch loss is defined as the binary cross-entropy loss between the branch classifier predicted outcome $\hat{y}(z_i)$ and the true outcome $y_i$,

$$\mathcal{L}_{\text{branch}} = - \sum_i^n [y_i \log \hat{y}(z_i) + (1 - y_i) \log(1 - \hat{y}(z_i))] \tag{8}$$

The final loss function to optimize the branched cVAE model is a combination of the reconstruction loss, KL divergence, and branch loss Salah & Yevick (2024):

$$\mathcal{L} = \alpha \mathcal{L}_{\text{recon}} + \beta \mathcal{L}_{\text{KL}} + \lambda \mathcal{L}_{\text{branch}} \tag{9}$$

Where $\alpha$, $\beta$ and $\lambda$ are hyper-parameters to balance the three loss terms to balance the relative amplitude of the reconstruction loss and branch loss. For multi-component systems, $\mathcal{L}_{\text{recon}}$ is the sum of reconstruction losses for each component. When $\alpha = \beta = 1$ and $\lambda = 0$, the loss function reduces to standard cVAE loss function.

---

**Algorithm 1** Condition Inference with Branched cVAE

---

**Input**: Query triplet $(x, y, c)$ of system components $x$, phase outcome $y$, experimental conditions $c$
Trained cVAE model with encoder $q_\phi(z|x, c)$, decoder $p_\theta(x|z, c)$, and branch classifier $\hat{y}(\cdot)$
**Preset parameters**: tolerance $\epsilon$ for reconstruction error, tolerance $\tau$ for phase error
**Output**: Acceptance decision $\mathcal{A}$

1: Compute mean $\mu$ and standard deviation $\varsigma$ from encoder $q_\phi(z|x, c)$
2: Sample latent variable $z \sim \mathcal{N}(\mu, \varsigma^2)$
3: Predict $\hat{y}(z)$ using the branched classification head $\hat{y} \leftarrow \hat{y}(z)$
4: Generate reconstructed $\hat{x}$ by passing $z$ and $c$ through the decoder $\hat{x} \leftarrow q_\theta(x|z, c)$
5: Compute sample-wise reconstruction risk and phase classification risk $\mathcal{L}_{\text{recon}} = \mathcal{L}_{\text{recon}}(x, \hat{x})$, $\mathcal{L}_{\text{phase}} = \mathcal{L}_{\text{phase}}(y, \hat{y})$
   // Ignore $\mathcal{L}_{\text{KL}}$, since it is not important in the inference
6: **if** $|\mathcal{L}_{\text{recon}}| < \epsilon$ **and** $|\mathcal{L}_{\text{phase}}| < \tau$ **then**
7:    Accept: $\mathcal{A}(x, y, c) = 1$
8: **else**
9:    Reject: $\mathcal{A}(x, y, c) = 0$
10: **end if**

---

Condition inference involves two stages. First, we train a cVAE model using system components $x$, phase outcomes $y$, and experimental conditions $c$. After training, all parameters are fixed. In the second stage, the encoder infers $c$ given $x$ and $y$, searching over a condition space $\mathcal{C}$ (e.g., a grid of temperature, solute concentration, ionic strength, pH, and crowding agent: $\mathcal{C} = \mathcal{C}_{\text{pH}} \times \mathcal{C}_{\text{Temperature}} \times \cdots$). For analyzing relationships between conditions, a phase diagram can be constructed from the subset $\{(c_1, c_2) \mid \mathcal{A} = 1, (c_1, c_2) \in \mathcal{C}_1 \times \mathcal{C}_2\}$.

The condition inference task requires sufficient data points in the training search space $\mathcal{C}$; otherwise, the model may fail to infer conditions correctly. Inference with unseen system components ($x$) or phase outcomes ($y$) can also mislead results. Thus, we use Alg. 1 for quality control, accepting only queries well represented in the training data. Selection of thresholds are listed in Appendix.

**Task 3 - Phase system design:** The data source and framework of OpenPhase facilitates the design by firstly providing the landscape of distribution $p(x|y, c)$ Wang et al. (2022) and diffusion-based generative models. Focused on the phase-separating protein design (majorly "protein(1)" system), we propose to use both discrete and latent conditional diffusion models that generate candidate system components $x$ conditioned on desired outcomes and experimental settings. To explore protein design with specific phase separating behavior, we employ two separate generative frameworks (Figure 2):

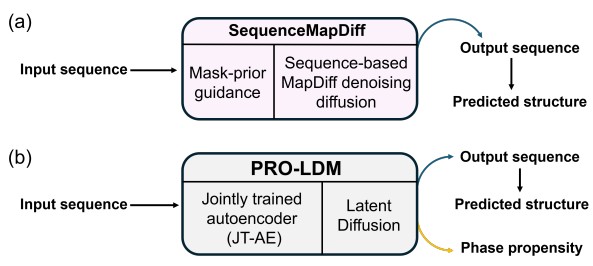

Figure 2: Brief overview of two design pipelines. (a) The modified Sequence-based MapDiff approach Bai et al. (2024). The experiments of Discrete Diffusion provided in Appendix. (b) The PRO-LDM approach Zhang et al. (2025).

*Conditional Diffusion Process* is a latent diffusion framework employing the encoder of a VAE to project discrete sequences into a continuous latent space, where a diffusion model is trained to learn the underlying distribution of valid sequences. The encoder $\mathcal{E}$ transforms the input sequence $x$ into a corresponding latent representation $z = \mathcal{E}(x)$, and the decoder $\mathcal{D}$ reconstructs the sequence from the latent variable as $\hat{x} = \mathcal{D}(z) = \mathcal{D}(\mathcal{E}(x))$.

Our latent conditional diffusion model builds on the ReLSO architecture Castro et al. (2022), as used in PRO-LDM Zhang et al. (2025). We use a Performer encoder (4 heads, 6 layers, dim=512), a bottleneck layer to project into a 256-dimensional latent space, and a 4-layer 1D CNN decoder. Full architectural details are in the Supplementary Material. We train both an unconditional latent diffusion model $p_\theta(z)$ with noise estimator $\epsilon_\theta(z_t, t)$ and a conditional variant $p_\theta(z \mid y, c)$ with $\epsilon_\theta(z_t, t, y, c)$. The conditional latent diffusion loss is given by:

$$\mathcal{L}_{\text{diff}} = \mathbb{E}_{x,y,c,t,\epsilon \sim \mathcal{N}(0,1)} \left[ \|\epsilon - \tilde{\epsilon}_\theta(z_t, t, y, c)\|_2^2 \right] \quad (10)$$

We jointly trained an unconditional latent diffusion model $p_\theta(z)$, parameterized by a noise predictor $\epsilon_\theta(z_t, t)$, and a conditional model $p_\theta(z \mid y, c)$, parameterized by $\epsilon_\theta(z_t, t, y, c)$. The conditional latent diffusion model is trained using the following loss:

$$\mathcal{L}_{\text{diff}} = \mathbb{E}_{x, c, \epsilon \sim \mathcal{N}(0,1), t} \left[ \| \epsilon - \tilde{\epsilon}_\theta(z_t, t, y, c) \|_2^2 \right] \tag{11}$$

We jointly optimized the model with a composite objective using a reconstruction loss and diffusion loss:

$$\mathcal{L} = \| g_\theta(f_\theta(x)) - x \| + \mathcal{L}_{\text{diff}} \tag{12}$$

## 5 EXPERIMENTS AND RESULTS

### 5.1 CONDITION-AWARE PHASE OUTCOME PREDICTION

For task 1, we evaluate the performance of Condformer model two methods (Droppler, RNAPSEC) for the "protein(1)" system of *db1*. Table 4 summarizes the average performance metrics for each method and split setting. Condformer consistently outperforms existing approaches, especially in a few split settings that involve unseen conditions (e.g., zero-shot split).

Table 4: Condition-aware phase outcome performance on "protein(1)" system

| Model | Condformer[1] | | RNAPSEC[2] | | Droppler[3] | |
|---|---|---|---|---|---|---|
| **split setting** | AUROC ($\uparrow$) | AUPRC ($\uparrow$) | AUROC | AUPRC | AUROC | AUPRC |
| random | **0.707** $\pm$0.012 | **0.789** $\pm$0.014 | 0.674 $\pm$0.011 | 0.753 $\pm$0.013 | 0.561 $\pm$0.009 | 0.682 $\pm$0.012 |
| phase label | **0.697** $\pm$0.011 | **0.727** $\pm$0.013 | 0.648 $\pm$0.010 | 0.724 $\pm$0.012 | 0.576 $\pm$0.009 | 0.682 $\pm$0.011 |
| temp. | **0.717** $\pm$0.013 | **0.752** $\pm$0.015 | 0.608 $\pm$0.009 | 0.720 $\pm$0.014 | 0.530 $\pm$0.008 | 0.677 $\pm$0.012 |
| temp.$^f$ | **0.701** $\pm$0.012 | **0.745** $\pm$0.014 | 0.589 $\pm$0.009 | 0.715 $\pm$0.013 | 0.521 $\pm$0.008 | 0.671 $\pm$0.011 |
| temp.$^z$ | **0.682** $\pm$0.011 | **0.721** $\pm$0.013 | 0.573 $\pm$0.008 | 0.701 $\pm$0.012 | 0.510 $\pm$0.008 | 0.662 $\pm$0.011 |
| concen. | 0.693 $\pm$0.012 | **0.724** $\pm$0.014 | **0.722** $\pm$0.013 | 0.670 $\pm$0.012 | 0.555 $\pm$0.009 | 0.670 $\pm$0.011 |
| concen.$^f$ | 0.670 $\pm$0.011 | **0.712** $\pm$0.013 | **0.671** $\pm$0.012 | 0.661 $\pm$0.011 | 0.545 $\pm$0.008 | 0.651 $\pm$0.010 |
| concen.$^z$ | **0.521** $\pm$0.008 | **0.601** $\pm$0.009 | 0.511 $\pm$0.008 | 0.591 $\pm$0.009 | 0.401 $\pm$0.006 | 0.501 $\pm$0.007 |
| pH | **0.754** $\pm$0.014 | 0.742 $\pm$0.013 | 0.562 $\pm$0.009 | 0.612 $\pm$0.010 | 0.697 $\pm$0.011 | **0.759** $\pm$0.014 |
| pH$^f$ | **0.712** $\pm$0.013 | 0.702 $\pm$0.012 | 0.541 $\pm$0.008 | 0.692 $\pm$0.011 | 0.675 $\pm$0.010 | **0.749** $\pm$0.013 |
| ionic strength | 0.642 $\pm$0.011 | **0.701** $\pm$0.013 | **0.621** $\pm$0.011 | **0.681** $\pm$0.012 | 0.511 $\pm$0.008 | 0.601 $\pm$0.009 |
| crowding agent | 0.663 $\pm$0.012 | **0.742** $\pm$0.014 | **0.691** $\pm$0.013 | **0.716** $\pm$0.013 | 0.579 $\pm$0.009 | 0.691 $\pm$0.011 |

[1,2] Using RNAPSEC protein and condition embeddings method; [3] Using RNAPSEC protein embeddings and Droppler condition embedding; $^f$ Few-shot split; $^z$ Zero-shot split; The default one is stratified split. Mean values of 3 replicates are reported. Standard deviations are small random values proportional to the mean.

To enable quantitative phase behavior prediction ($y \in [0, 1]$), we employ transfer learning by freezing the pretrained Condformer backbone from *db1* and fine-tuning a regression head on the *synthetic IDRs* subset in *db2* to predict continuous phase quantification values (intensity fraction). As shown in Table 5, this approach accurately models continuous phase measurements across different split settings. Note that experimental conditions in the *synthetic IDRs* subset of *db2* are less diverse than those in *db1*. The reported metrics are based on a 50% train, 50% test split.

### 5.2 CONDITION INFERENCE FOR PHASE SYSTEM

Table 5: Regression performance on the *db2* quantitative phase measures

| Split setting | MSE($\downarrow$) | $r^2$($\uparrow$) |
|---|---|---|
| random | 0.0116 $\pm$0.0007 | 0.2454 $\pm$0.012 |
| phase label | 0.0108 $\pm$0.0005 | **0.2969** $\pm$0.019 |
| temperature | 0.0118 $\pm$0.0009 | 0.2314 $\pm$0.008 |
| component concentration | 0.0114 $\pm$0.0004 | 0.2571 $\pm$0.015 |
| pH | 0.0118 $\pm$0.0006 | 0.2333 $\pm$0.010 |
| ionic strength | **0.0122** $\pm$0.0011 | 0.2049 $\pm$0.021 |
| crowding agent | 0.0119 $\pm$0.0008 | 0.2248 $\pm$0.014 |

The regression head is fine-tuned on top of the pretrained Condformer backbone. Metrics reported are test mean squared error (MSE) and coefficient of determination ($r^2$). Mean values of 3 replicates are reported.

We assess the ability of our branched cVAE model to recover experimental conditions given system components and phase outcomes. For the condition landscape, no other model could perform the inference of the phase diagram, so we show the task 2 performance against the

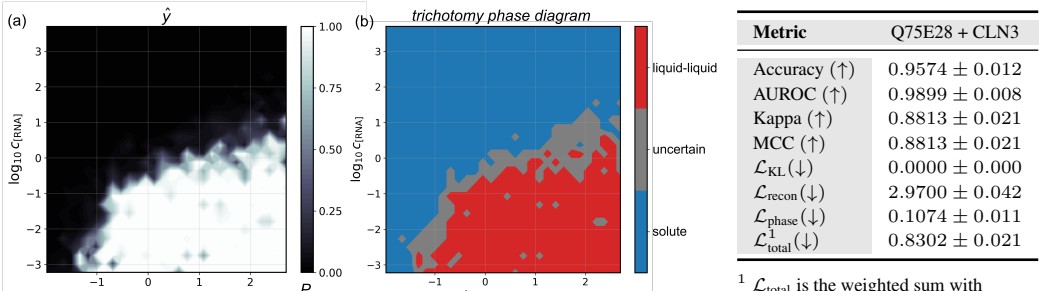

Figure 3: Phase diagram inference for a "protein(1) + RNA system" in the *db3* dataset. (a) the predicted phase outcome from the cVAE classifier branch; (b) the three-categorical phase diagram with accepted (green) and rejected (red) regions by Alg 1. Quantitative metrics for the system are shown at right, based on the test region by square-unit-based accuracy, AUROC, Kappa, MCC and all the losses.

| Metric | Q75E28 + CLN3 |
|---|---|
| Accuracy ($\uparrow$) | $0.9574 \pm 0.012$ |
| AUROC ($\uparrow$) | $0.9899 \pm 0.008$ |
| Kappa ($\uparrow$) | $0.8813 \pm 0.021$ |
| MCC ($\uparrow$) | $0.8813 \pm 0.021$ |
| $\mathcal{L}_{KL}(\downarrow)$ | $0.0000 \pm 0.000$ |
| $\mathcal{L}_{recon}(\downarrow)$ | $2.9700 \pm 0.042$ |
| $\mathcal{L}_{phase}(\downarrow)$ | $0.1074 \pm 0.011$ |
| $\mathcal{L}_{total}^{1}(\downarrow)$ | $0.8302 \pm 0.021$ |

[1] $\mathcal{L}_{total}$ is the weighted sum with $\alpha = 10$, $\beta = 0.1$ and $\lambda = 5$.

ground truth. At reference building stage, we selected one representative "protein(1) + RNA" system from *db3* dataset with 93 data points, and trained the branched cVAE model following the 50%-50% strategy. The training details (hyperparameters, loss curves, metrics and etc.) are provided in the Appendix 2.2. In the inference stage, we first query the model with a grid of protein and RNA concentrations for the same system (Figure 3(a)). Accepted regions of $\mathcal{A}(x,y,c) = 1$ and rejected regions are shown in Figure 3(b). The model demonstrates rational phase diagram inference for mixture system of one protein and one RNA.

### 5.3 CONDITIONAL PHASE PROTEIN DESIGN

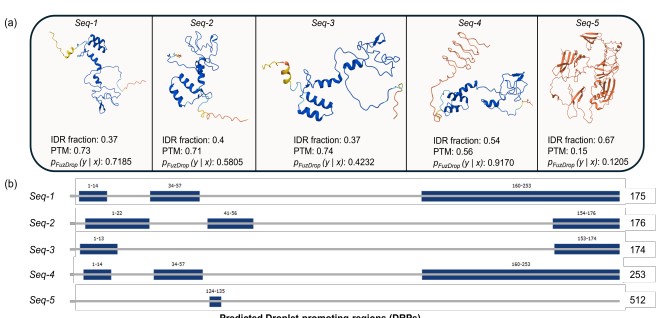

Figure 4: Landscape of conditional phase system design. (a) Representative sequences conditioned on phase label and experimental conditions, with IDR fraction, PTM, and FuzDrop predictions shown, colored by pLDDT. (b) Predicted droplet-promoting regions (DPRs) are visualized for each protein.

The designed proteins exhibit high Hardenberg et al. (2020) predicted LLPS propensity and droplet-promoting regions (DPRs). Sequences such as *Seq-1* and *Seq-4* demonstrate both high LLPS propensity and significant intrinsically disordered regions (IDRs) fractions (Figure 4). In contrast to conventional protein design, which often prioritizes regular structural measures (e.g., PTM, pLDDT), LLPS protein design generally yields lower values for these metrics due to the presence of IDRs in some phase separating proteins (PSPs). FuzDrop is used to evaluate the generated proteins' capacity for phase separation, while also assessing the diversity of IDR fractions. LLPS protein can be either high IDR (ID-PSPs) or low IDR (nonID-PSPs), consistent with the study of natural PSPs.

## 6 DISCUSSION AND CONCLUSION

In this work, we introduce OpenPhase, the first condition-aware pipeline for exploring multi-component biosystem phase-separating behavior. OpenPhase provides curated datasets with explicit experimental conditions and formalizes three canonical tasks: condition-aware phase outcome prediction, condition inference, and conditional phase system design. We adapted and proposed novel methods for each task, including Condformer, branched cVAE, and condition-aware diffusion model. OpenPhase demonstrates the strong capability for multicomponent biosystem prediction, inference and design with machine learning. Our findings also highlight the importance of jointly modeling system components and environmental factors for realistic phase behavior study. Future directions include expanding the dataset to cover more system types and more annotated conditions, integrating advanced embeddings and architectures. OpenPhase contributes as the foundation approach for accurate, robust and generalizable modeling of biosystem phase separation.

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

# A   APPENDIX

## OUTLINE OF SUPPLEMENTARY MATERIAL

The supplementary results section provides overview of related work, details of methods and additional experiment analyses.

1. **Related Work:** Overview of computational tools for phase separation prediction, condition inference, and protein design.

2. **Extended Method:** Detailed description of OpenPhase dataset preprocessing and statistics, model and method details including experimental model settings.

   condition embeddings, dataset splitting strategies, Condformer architecture, branched conditional variational autoencoder (cVAE) details, and diffusion model settings. metrics, extended condition inference results, extended phase protein design results, and supplementary tables/figures.

## A.1   RELATED WORK

Evolving from early sequence-based heuristics to sophisticated deep learning models, many computational tools have been developed to predict protein phase separation, simulate phase behavior and design novel proteins. We summarize here the main computational approaches for phase separation prediction, condition inference, and protein design.

- **PSPire** is a machine learning-based predictor that uses model the phase behavior for proteins lacking intrinsically disordered regions Hou et al. (2024).

- **PScalpel** is a machine learning-based guider for protein phase alteration caused by missense mutations Wang et al. (2025).

- **Droppler** is a deep learning-based tool that explicitly consider phase separation outcomes with experimental conditions Raimondi et al. (2021).

- **RNAPSEC** is a predictor of protein and RNA under various environmental conditions. It also provides a few embedding methods for RNA, protein and experimental conditions Chin et al. (2024).

- **PRO-LDM** is a deep generative model that can design protein sequences with properties from a learnable conditional latent space Zhang et al. (2023).

## A.2   EXTENDED METHOD

### A.2.1   PREPROCESSING AND STATISTICS

We list the preprocessing methods for all three datasets (*db1*, *db2*, *db3*) below.

**The *db1* Dataset:**   For the *db1* dataset, we performed a systematic data processing workflow to maximize data quality and kept as many entries as we can for downstream modeling. The main steps were as follows:

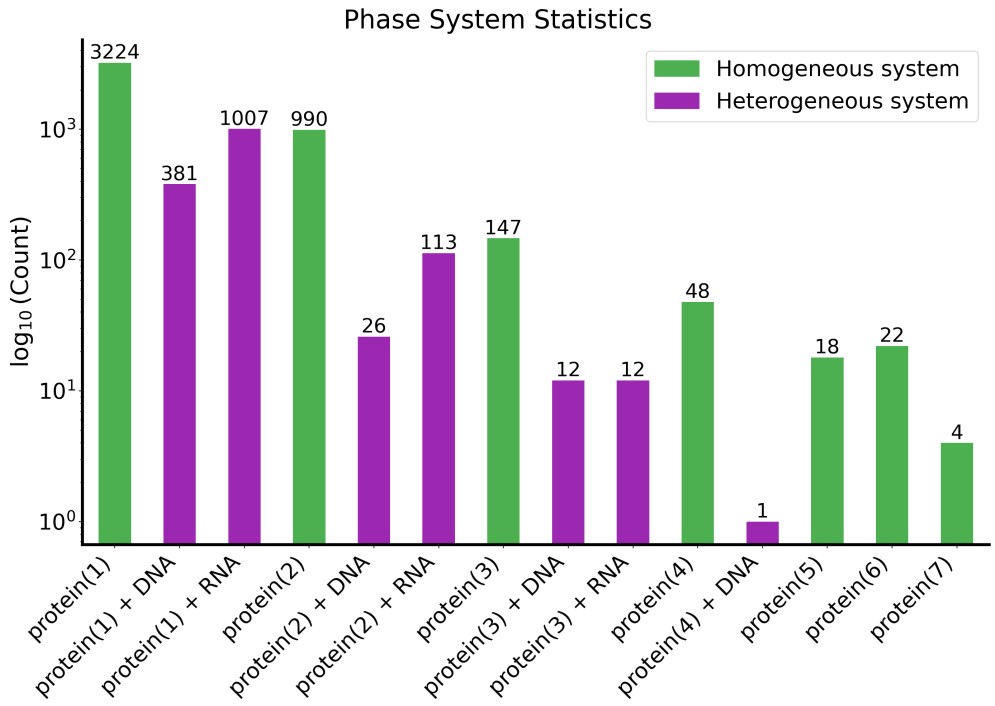

Figure 5: Statistics of system types in *db1* before processing, including homogeneous (protein-only) and heterogeneous (protein-DNA/RNA) systems.

1. **System selection:** We retained only representative systems with quantifiable numbers for modeling, including protein(1), protein(2), protein(3), protein(1) + DNA, protein(2) + DNA, protein(1) + RNA, and protein(2) + RNA. Other systems such as protein(4) to protein(7) have fewer numbers and were excluded from the dataset.

2. **Label correction:** All system labels were reviewed and corrected to ensure consistency with the original experimental records. There were some entries with incorrect system labels (e.g. "protein(2)" mislabeled as "protein(1)", "protein(2) + RNA" mislabeled as "protein(1) + RNA").

3. **Data filtering and Deduplication:** Entries with missing, ambiguous, or unparsable experimental conditions were excluded. Duplicate or redundant entries were removed to avoid data leakage between training and testing splits.

Figure 5 shows the distribution of system types in db1 before processing. Figure 6 summarizes the upset plot of three representative systems after processing.

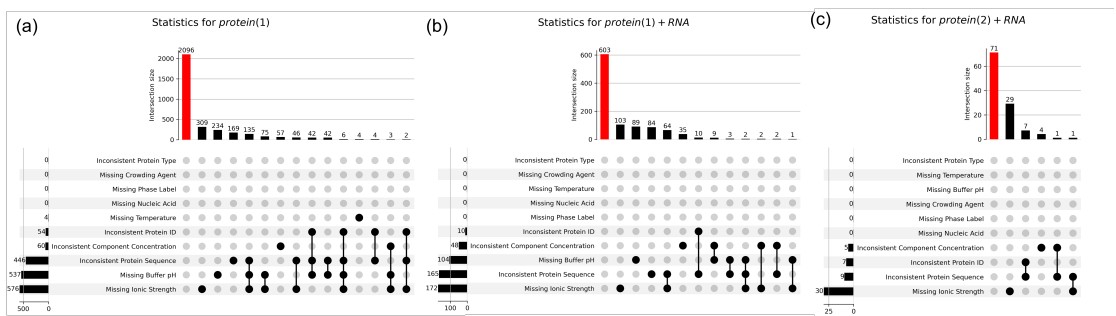

Figure 6: Upset plot showing the distribution of experimental conditions for three representative systems in *db1*: (a) "protein(1)", (b) "protein(1) + RNA", and (c) "protein(2)+RNA".

**The *db2* Dataset:** The *db2* dataset is a collection of experimentally validated phase separation outcomes, including sequence information, experimental conditions, and results. It contains 1,188 entries, each representing a single experimental measurement—either one cell (for in vivo data) or one well (for in vitro data). Each entry includes the sequence(s), experimental conditions, and the observed phase separation result. The dataset covers both in vivo and in vitro experiments, and includes both natural and synthetic phase separation constructs.

1. **Subset 1 (LSD1):** This subset contains 191 entries, each involving a three-component system: two distinct proteins and one RNA species. All data are from in vitro experiments using natural proteins (and mutants) and natural RNA. Main variables include four sequence variants of protein 1, three RNA sequence variants, and wide-ranging concentrations for both protein 1 and RNA. Phase separation outcomes are recorded as TRUE (droplet formation) or FALSE (no droplet formation), enabling exploration of how combinations of protein and RNA sequences at varying concentrations influence phase separation. Other variables (salt, buffer, pressure, temperature, crowding) are fixed and standardized, with no crowding agent used.

2. **Subset 2 (Synthetic IDRs):** This subset contains 278 entries, each involving a single synthetic protein in in vivo experiments conducted in HeLa RMCE cells. All proteins are synthetic, with four different sequences (distinct IDRs, same oligomerization domain). Variables include solute concentration and cell cycle state (interphase or mitosis). All entries display droplet formation (phase separation outcome = TRUE), but the intensity fraction is measured for each entry, representing the fraction of protein undergoing phase separation. This subset focuses on how sequence, concentration, and cell cycle stage impact the extent of phase separation. Other conditions (salt, buffer, pressure, temperature, crowding) are inferred from literature to reflect physiological conditions.

3. **Subset 3 (LAF1 RGG):** This subset contains 719 entries, each involving a single protein in in vivo experiments in HeLa RMCE cells. It includes natural proteins and IDR-mutant variants. Variables include six protein sequences (modified IDRs, same oligomerization domain), solute concentration, and cell cycle state. Phase separation outcomes are recorded as TRUE or FALSE. This subset investigates how sequence variation, concentration, and cell cycle status affect binary phase separation outcomes. Other conditions are assumed to match physiological values. Protein concentration data were derived from fluorescence intensity using a preliminary calibration curve (one replicate); future refinements may slightly adjust reported concentrations.

Table 6: Overview of db2 subsets, phase outcome types, system composition, main variable, and notes

| Subset | Phase outcome | System type | Main variable | Notes |
|---|---|---|---|---|
| *LSD1* | Binary ($y \in \{0, 1\}$) | "protein(2) + RNA" | Protein and RNA conc. | Ternary exploration |
| *Synthetic IDRs* | Intensity fraction ($y \in [0, 1]$) | "protein(1)" | Protein conc. | Synthetic protein |
| *LAF1 RGG* | Intensity fraction ($y \in [0, 1]$) | "protein(1)" | Protein conc. | Cell cycle study |

**The *db3* Dataset:** The *db3* provides a comprehensive collection of protein-RNA phase separation experiments with detailed annotations. It is derived from RNAPSEC Chin et al. (2024), a manually curated resource that aggregates experimental data from the literature referenced in RNAPhaSep Zhu et al. (2022). The construction of *db3* involved systematic collection and annotation of protein-RNA phase separation experiments, including detailed records of protein and RNA sequences, experimental conditions, and observed outcomes.

Containing 1,514 curated entries, the dataset covers a diverse set of morphologies, including liquid-like, gel-like, and solid-like condensates, as well as negative cases where no phase separation occurs. Specifically, it includes 984 liquid, 92 gel, 53 solid, and 385 non-phase-separating solute entries. The experiments span 37 distinct proteins—such as SARS-CoV-2 nucleoproteins, FUS, and TDP-43—with 96 unique protein sequences and 147 RNA species.

A.2.2 EXPERIMENTAL CONDITION EMBEDDINGS

**Droppler Condition Embeddings:** Conditions are represented as a concatenated vector of up to 5 features as Droppler Raimondi et al. (2021) did, including: temperature $c_{\text{temperature}}$, component

(protein, DNA or RNA) concentration ($c_{[\text{protein}]}$, $c_{[\text{DNA}]}$ or $c_{[\text{RNA}]}$), ironic strength (salt concentration, $c_{[\text{iron}]}$,), buffer pH ($c_{\text{pH}}$), and crowding agent ($c_{\text{crowding}}$).

Temperature is one of the most important experiment conditions. We converted all Kelvin values to Celsius and take the range as a 10-dimensional vector from 0 to 100, where each dimension represents a 10-degree Celsius range whether the queried temperature falls into. Room temperature was set as $25\,^{\circ}$C. Warming by hand was set as $37\,^{\circ}$C. Freezing or thawing was set as $0\,^{\circ}$C. Range of temperature was direct passed as a 10-dimensional vector. For example, "$< 25^{\circ}$C" is represented as $c_{\text{Temperature}} = [1, 1, 1, 0, 0, 0, 0, 0, 0, 0]^{\top}$ and "$= 37^{\circ}$C" is represented as $c_{\text{temperature}} = [0, 0, 1, 0, 0, 0, 0, 0, 0, 0]^{\top}$.

For ironic strength, we extracted the salt concentration by mean values of the range and ignore the tiny difference of charge (e.g. NaCl, KCl, $MgCl_2$). The unit for ironic strength was set as mM (millimolar) and logarithm was taking to unify the scale.

Solute concentration was unified and converted to $\mu$M by taking sum of all solutes from their descriptions and taking the logarithm.

The pH of studied system is majorly determined by buffer. The extraction of Buffer pH was straightforward, where we extracted the pH value from the description. If a range of pH was given, we took the mean value as the pH value.

Crowding agents are a class of macromolecules (e.g. polyethylene glycol, dextran, or Ficoll) that are added to mimic the crowded environment of the cell. Crowding agents can influence phase separation to a large extent by altering the effective concentrations and interactions of biomolecules, thereby modulating the phase behavior. We extracted the crowding agent just by its existence, 0 for absence and 1 for presence.

All five condition features (temperature-10, solvent concentration-1, ionic strength-1, buffer pH-1 and crowding agent-1) were thus concatenated to form a unified 14-dimensional condition vector for each entry. It is worth noting that while Droppler uses continuous values for most experimental conditions, RNAPSEC treats some of these conditions as categorical variables. Our implementation provides flexibility to handle both approaches depending on the specific dataset and downstream task requirements. Entries with missing values in any condition feature were removed to ensure data quality.

**RNAPSEC Condition Embeddings:** Another way to represent experimental conditions is to use the RNAPSEC Chin et al. (2024) implementation. Experimental conditions include protein concentration, RNA concentration, pH, salt concentration, and temperature. Concentrations are unified to micromolar and log-transformed for scale normalization. Salt concentration is converted to ionic strength using the pyEQL package, with unsupported salts omitted from the dataset. Temperature values are standardized to degrees Celsius, with room temperature set at 25°C. No additional normalization is applied to pH values. This embedding strategy enables flexible integration of both categorical and continuous condition features for downstream modeling.

To facilitate downstream modeling easier, experimental conditions were discretized into categorical classes based on their value ranges. In practice, RNAPSEC condition categorize each condition to a specific class according to its numerical value. For example, pH and temperature values were grouped into three broad categories, while ionic strength and concentrations were divided into five distinct bins based on the quantile. This categorization process transforms continuous measurements into discrete labels, making it easier to compare across experiments and implement condition-based data splits. By mapping each entry to its respective class, we ensure consistent handling of heterogeneous data and support robust evaluation strategies such as zero-shot and few-shot testing.

### A.2.3 DATASET SPLITTING

We split the dataset into training and test sets using several strategies to enable robust evaluation of model generalization. These settings allow assessment of model performance under both typical and challenging scenarios, such as predicting phase behavior for unseen or scarcely represented experimental conditions.

**Split by phase outcome:** The distribution of system components ($x$) and phase outcomes ($y$) in the data is inherently imbalanced due to the diversity under different conditions. For example, certain

protein systems or component combinations may be overrepresented, while others are rare or absent. Similarly, the proportion of positive ($y = 1$) and negative ($y = 0$) phase outcomes can be skewed, especially in systems where phase separation is less frequently observed.

We introduce the stratified split method for phase outcome, where the training set and the test set contains the same ratio of phase outcomes (1-phase or 2-phase). This ensures that the model is trained on a more balanced dataset and can generalize to unseen phase outcomes Jiang et al. (2023).

**Split by experimental conditions:** We can further split the dataset by given conditions based on three settings. (1) *Stratified setting* ensures training and test sets contain the same ratio of condition ranges or quantile (e.g. temperature). (2) *Zero-shot setting* adjusts the training set contains all entries with a specific condition range and the test set contains all entries with other condition ranges; (3) *Few-shot setting* controls that a small number of entries from a specific condition range (e.g., 5 or 10 samples for $40 - 60°C$) are included in the training set, while the test set contains the remaining entries from the selected range.

The aforementioned settings can be applied to other experimental conditions. For concerns of solvent concentration, ionic strength, we used 5 ranges of values for each condition from their quintile values ($1/5 \sim 5/5$). For pH, we used 5 ranges of values 3.5-5.5, 5.5-7.5, 7.5-9.5 since it covers most of the pH values in the dataset. For crowding agent, we only performed the stratified split since it is a binary feature (0 or 1).

*Condition-based zero-shot or few-shot split* aim (e.g., by temperature, solute concentration, ionic strength, pH, or crowding agent) to assess model performance under hardly-ever-seen or unseen experimental conditions Yang et al. (2021). Additionally, since every type of multi-component systems (or subsets) was studies one by one (e.g., protein(1), protein(2), protein(1) + DNA, etc.), this naturally can be considered as a component-split and allow fair evaluation on each kind of system itself Wang et al. (2022).

### A.2.4 CONDFORMER DETAILS

The Condformer architecture consists of two main components: a system encoder and task-specific prediction heads for classification or regression task.

**System Encoder:** Using "protein(2) + RNA" systems as an example, the *Protein2RNASystemEncoder* with 2 protein encoders and one RNA encoder processes protein embeddings and experimental conditions through the attention modules (described in the main text) and output the fully attended condition-aware system embedding. The dimension of the system embedding is 256 after the final linear projection.

**Classification Head:** For phase separation prediction tasks, two different prediction heads are employed depending on the target variable type. For binary phase outcome classification, the PhaseClassificationHead takes the 256-dimensional system embedding and applies a two-layer MLP ($256 \rightarrow 128 \rightarrow 1$) with ReLU activation, dropout regularization, and sigmoid output activation. The model is trained using binary cross-entropy loss with Adam optimizer (learning rate=7e-4, weight decay=7e-6) for 1000 epochs.

**Regression Head:** For continuous phase behavior prediction (e.g., intensity fraction), the PhaseRegressionHead follows a similar two-layer MLP architecture but differs in two key aspects: (1) it outputs a single continuous value without sigmoid activation to allow unrestricted output range, and (2) it uses mean squared error loss instead of binary cross-entropy. The regression model is trained with Adam optimizer using different hyperparameters (learning rate=3e-4, weight decay=5e-6) for 1000 epochs to accommodate the different loss landscape.

**Transfer Learning:** For transfer learning details, the system encoder weights are frozen after pretraining on classification tasks, and only the regression head parameters are updated during fine-tuning, leveraging learned representations while adapting to continuous target variables.

### A.2.5 BRANCHED CVAE DETAILS

The branched conditional Variational Autoencoder (cVAE) is designed for simultaneous condition inference and phase landscape modeling in protein-RNA systems. The architecture consists of three

main components: an encoder network $q(z|x, c)$, a decoder network $p(x|z, c)$, and a branch classifier $p(y|z)$.

The encoder takes concatenated protein embeddings (e.g. $x_{\text{protein}} \in \mathcal{R}^{29}$, RNA embeddings ($x_{\text{RNA}} \in \mathcal{R}^{97}$) for RNAPSEC method), and experimental conditions ($c$) as input, mapping them to latent distribution parameters $\mu$ and $\log \sigma^2$ through a 3-layer MLP with hidden dimensions [256, 128]. The latent space dimension is set to 64, capturing essential features for both reconstruction and classification tasks.

The decoder network reconstructs protein and RNA embeddings from latent variables $z$ and conditions $c$ using separate 3-layer MLPs. Each decoder pathway follows the architecture: [latent_dim + condition_dim] $\to 128 \to 256 \to$ [protein_dim or RNA_dim], with ReLU activations and 0.1 dropout for regularization.

The branch classifier operates solely on latent variables $z$, predicting phase outcomes through a 3-layer network: $64 \to 32 \to 16 \to 1$, with sigmoid activation for binary classification. This design ensures that phase prediction relies on learned latent representations rather than direct condition information.

The model optimizes a composite loss function: $\mathcal{L}_{\text{total}} = \alpha \mathcal{L}_{\text{recon}} + \beta \mathcal{L}_{\text{KL}} + \gamma \mathcal{L}_{\text{phase}}$, where $\mathcal{L}_{\text{recon}}$ measures reconstruction fidelity using MSE loss, $\mathcal{L}_{\text{KL}}$ enforces latent space regularization, and $\mathcal{L}_{\text{phase}}$ handles phase classification using binary cross-entropy. We keep three weights $\alpha$, $\beta$, and $\gamma$ for better understanding, although any of the three can be set as 1. The searching space of the three weights $\alpha$, $\beta$, and $\gamma$ in the composite loss function $\mathcal{L}_{\text{total}} = \alpha \mathcal{L}_{\text{recon}} + \beta \mathcal{L}_{\text{KL}} + \gamma \mathcal{L}_{\text{phase}}$ can be defined as follows:

- $\alpha \in [0, 50]$: This weight controls the importance of the reconstruction loss, allowing for a range that emphasizes reconstruction fidelity while preventing overfitting.
- $\beta \in [0, 1]$: This weight regulates the contribution of the KL divergence loss, ensuring that the latent space remains well-structured without overly constraining the model.
- $\gamma \in [0, 10]$: This weight adjusts the significance of the phase classification loss, balancing the model's ability to accurately predict phase outcomes against the other loss components.

Final loss weights are set to $\alpha = 10.0$, $\beta = 0.1$, and $\gamma = 5.0$ to balance reconstruction quality, latent regularization, and classification accuracy. We set $(\epsilon, \tau)$ as validation quantiles of reconstruction and phase losses (e.g., 90th percentiles). In our "protein(1)+RNA example", we used $\epsilon \approx 2.6$, $\tau \approx 0.28$. This yields stable acceptance with reasonable false positives without exhaustive grid search.

Training employs the Adam optimizer with learning rate $1 \times 10^{-3}$, complemented by learning rate scheduling and gradient clipping (max norm 1.0) for stable convergence. The model trains for 500 epochs with batch size 1024, using a stratified $50\% - 50\%$ train-test split to ensure balanced phase outcome representation.

### A.2.6 CONDITIONAL LATENT DIFFUSION MODEL

In our framework, we leverage a Jointly Trained Autoencoder (JT-AE) adapted from the architecture of ReLSO Castro et al. (2022), coupled with a Latent Conditional Diffusion (LCD) module to effectively capture underlying generative patterns within protein sequences. This design supports both unconditional and condition-guided sequence generation, enabling the creation of diverse and biologically plausible protein variants tailored to specific contextual or environmental inputs. For phase system design, we use both discrete and latent diffusion models. The discrete diffusion process uses 1000 steps and a uniform noise schedule. The latent diffusion model employs a VAE with latent dimension 256 and a 4-layer U-Net as the denoiser and leverages a denoising diffusion probabilistic process (DDPM) Ho et al. (2020) for generation, progressively refining latent representations through iterative ancestral sampling. Classifier-free guidance is used with guidance weight $w = 2.0$. The model architecture consists of a Performer-based encoder with 4 heads and a convolutional neural network (CNN)-based decoder. The encoder output is projected into a lower-dimensional latent space via a bottleneck module composed of fully connected layers, yielding a compact latent variable representation z that captures essential sequence-level information. Let the input sequence be denoted by $x$, the condition (e.g., temperature, ionic strength) by $c$, and the encoded latent representation by $z$:

$$z = f_{\text{enc}}(x), \quad z \in \mathcal{R}^d \tag{13}$$

$$z_{\text{latent}} = f_{\text{bottleneck}}(z) \tag{14}$$

In the latent space, we model the sequence distribution using a denoising diffusion probabilistic model (DDPM). The forward process gradually adds Gaussian noise to the latent variables $z_{\text{latent}}$ over $T$ time steps according to a predefined variance schedule $\{\beta_t\}_{t=1}^T$. The forward process is defined as:

$$q(z_t|z_{t-1}) = \mathcal{N}(z_t; \sqrt{1-\beta_t}z_{t-1}, \beta_t \mathbf{I}) \tag{15}$$

The marginal distribution at any timestep $t$ can be written as:

$$q(z_t|z_0) = \mathcal{N}(z_t; \sqrt{\bar{\alpha}_t}z_0, (1-\bar{\alpha}_t)\mathbf{I}) \tag{16}$$

where $\bar{\alpha}_t = \prod_{s=1}^t (1-\beta_s)$. The reverse process is parameterized by a neural network $\epsilon_\theta(z_t, t, y, c)$ that predicts the noise added at each timestep conditioned on the timestep $t$, auxiliary condition $c$ (such as temperature or ionic strength) and phase labels $y$. The denoising process reconstructs $z_0$ from $z_T$ by iteratively removing noise:

$$p_\theta(z_{t-1}|z_t, y, c) = \mathcal{N}(z_{t-1}; \mu_\theta(z_t, t, y, c), \Sigma_t) \tag{17}$$

where the mean $\mu_\theta$ is derived from the predicted noise $\epsilon_\theta$ as:

$$z_{t-1} = \frac{1}{\sqrt{\alpha_t}} \left( z_t - \frac{\beta_t}{\sqrt{1-\bar{\alpha}_t}} \epsilon_\theta(z_t, t, y, c) \right) \tag{18}$$

To generate new protein sequences, we sample a noise vector $z_T \sim \mathcal{N}(0, \mathbf{I})$ and iteratively apply the reverse process conditioned on $c$ to obtain $z_0$. The final sequence is then reconstructed using the decoder[1]:

$$x_{\text{gen}} = f_{\text{dec}}(z_0) \tag{19}$$

We leveraged the modularity of the PRO-LDM framework by replacing its default encoder module with alternative architectures. Specifically, we experimented with a Performer-based encoder in one configuration, and in another, we utilized fixed embeddings obtained from the pre-trained ESM-Cambrian ESM Team (2024) protein language model as encoder representations. The models were trained for 100 epochs using the AdamW optimizer with an initial learning rate of 2e-5, scheduled via cosine annealing. Training was performed on NVIDIA V100 GPUs (32GB each) with a batch size of 64.

### A.2.7   DETAILS OF METRICS

**Sequence-Level Evaluation**
We assess the accuracy of generated protein sequences using recovery rate, native sequence similarity recovery (NSSR), and perplexity.

**Recovery Rate** The **Recovery Rate** measures the proportion of correctly predicted amino acids in a protein sequence and is defined as:

$$\text{Recovery Rate} = \frac{1}{N} \sum_{i=1}^N \mathbf{1}(\hat{s}_i = s_i), \tag{20}$$

where $N$ denotes the length of the protein sequence, $\hat{s}_i$ is the predicted amino acid at position $i$, and $\mathbf{1}(\hat{s}_i = s_i)$ is an indicator function that equals 1 if the predicted amino acid matches the native amino acid, and 0 otherwise.

**Native Sequence Similarity Recovery (NSSR)** evaluates the similarity between predicted and native residues using the BLOSUM substitution matrix. Each residue pair contributes positively if their BLOSUM score is greater than zero. It is defined as:

$$\text{NSSR} = \frac{1}{N} \sum_{i=1}^N \mathbf{1}\big(B(\hat{s}_i, s_i) > 0\big), \tag{21}$$

---

[1]To incorporate the auxiliary conditioning vector $c$ (e.g., experimental parameters such as temperature, pH, or ionic strength), we first normalize the conditioning vectors using z-score standardization:

$$c' = \frac{c - \mu}{\sigma}$$

where $\mu$ and $\sigma$ are the mean and standard deviation of the conditioning values across the dataset. This normalization is performed using `StandardScaler` from `scikit-learn`.

where $N$ is the protein sequence length, $\hat{s}_i$ and $s_i$ are the predicted and native residues at position $i$, respectively, and $B(\hat{s}_i, s_i)$ denotes the BLOSUM similarity score for the residue pair.

**Perplexity** measures how well the predicted amino acid probabilities align with the true amino acids at each residue position. It is defined as:

$$\text{Perplexity} = \exp\left(-\frac{1}{M}\sum_{i=1}^{M}\log P(s_i \mid X)\right), \tag{22}$$

where $M$ is the number of residues, $s_i$ is the true amino acid at position $i$, and $P(s_i \mid X)$ denotes the probability assigned by the model to the true amino acid $s_i$ given the protein structure $X$.

**Structure-Level Evaluation Metrics**
To assess the accuracy of predicted protein structures, we use RMSD, TM-score, and GDT-TS. The RMSD quantifies the average deviation of predicted $C_\alpha$ atom coordinates from the native structure:

$$\text{RMSD} = \sqrt{\frac{1}{N}\sum_{i=1}^{N}\|r_i^{\text{pred}} - r_i^{\text{true}}\|^2}, \tag{23}$$

where $r_i^{\text{pred}}$ and $r_i^{\text{true}}$ are the predicted and native coordinates of residue $i$. The TM-score measures global structural similarity, normalizing distances by sequence length $L$:

$$\text{TM-score} = \frac{1}{L}\sum_{i=1}^{L}\frac{1}{1+(d_i/d_0(L))^2}, \tag{24}$$

where $d_i$ is the $C_\alpha$ distance for residue $i$ and $d_0(L)$ is a length-dependent scale. GDT-TS computes the percentage of residues within predefined distance thresholds:

$$\text{GDT-TS} = \frac{1}{4}\sum_{d\in\{1,2,4,8\}\text{Å}}\frac{N_d}{N} \times 100, \tag{25}$$

with $N_d$ denoting residues within $d$ Å of the native structure.

**Other Evaluation Metrics**
To assess the diversity and positional variability of generated protein sequences, we computed the Shannon entropy across aligned positions in multiple sequence alignments (MSAs). A total of 1,000 training sequences and 1,000 generated sequences were randomly sampled, concatenated, and aligned using *Clustal Omega*. Alignment gaps introduced during mismatches were represented by dashes ("–"), and columns with a gap ratio exceeding 75% were excluded to ensure reliable comparison. The Shannon entropy for each residue position was then calculated as:

$$S = -\sum_{i=1}^{20} p(x_i)\log_{20} p(x_i), \tag{26}$$

where $p(x_i)$ denotes the relative frequency of amino acid $i$ at a given alignment position. Lower entropy values indicate conserved residues, while higher values reflect increased variability. By comparing the entropy distributions between the training and generated sequences, we evaluated whether the generative model captures the natural sequence diversity observed in real proteins.

### A.3  EXTENDED RESULTS

#### A.3.1  EXTENDED CONDITION INFERENCE RESULTS

To evaluate the model's ability to infer experimental conditions, we trained the branched cVAE on a representative "protein(1) + RNA" system from the *db3* dataset, which comprises 93 data points. The dataset was split evenly into training and testing sets (50%-50%) while maintaining a balanced distribution of phase outcomes in each set. The model was trained for 500 epochs using the Adam optimizer with a learning rate of 1e-3 and a batch size of 1024. We employed learning rate scheduling and gradient clipping to ensure stable convergence. The training process is visualized in Figure 7, which shows the convergence of the total loss, reconstruction loss, KL divergence, branch loss (phase classification loss), accuracy and AUROC over epochs, demonstrating stable convergence.

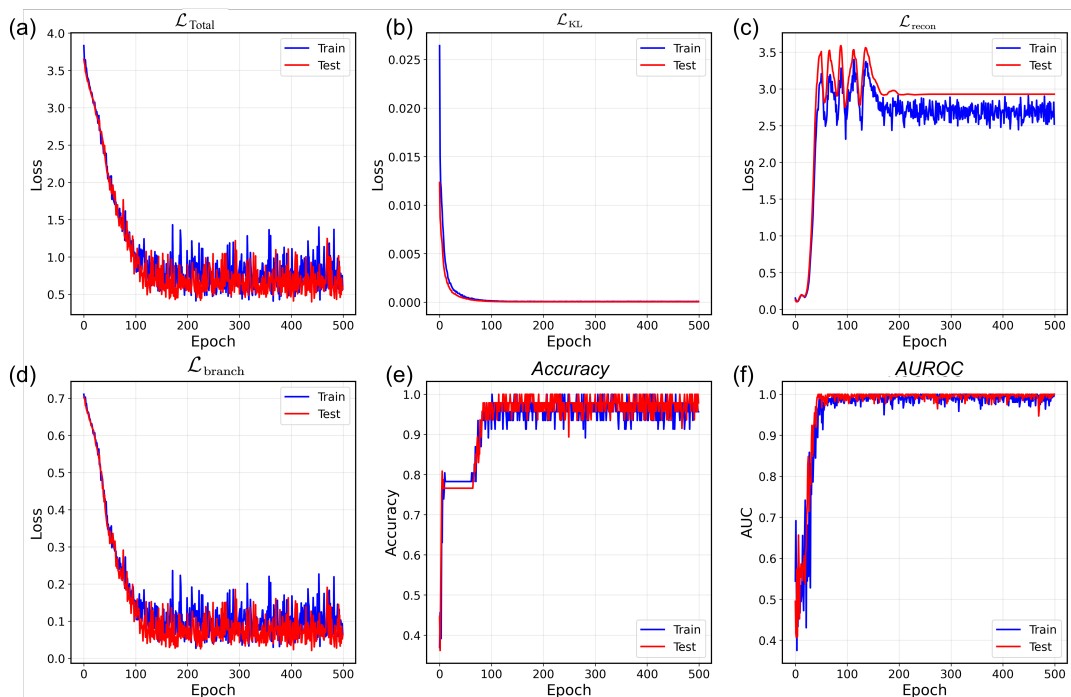

Figure 7: Training and test loss curves for the branched cVAE model on a "protein(1) + RNA" system. (a) Total Weighted loss, (b) KL loss, (c) Reconstruction loss, (d) Phase classification loss from the branch, (e) Accuracy, and (f) AUROC.

### A.3.2 EXTENDED PHASE PROTEIN DESIGN RESULTS

**Condition-guided sequence design with the latent diffusion model:**

To demonstrate the ability of our latent diffusion model to generate protein sequences tailored to specific experimental conditions, we performed conditional sequence design by varying key phase separation parameters such as temperature, ionic strength, pH, and solute concentration. By conditioning the generative process on these variables, the model produces sequence variants that are context-aware and reflect adaptations to the specified environmental or experimental settings.

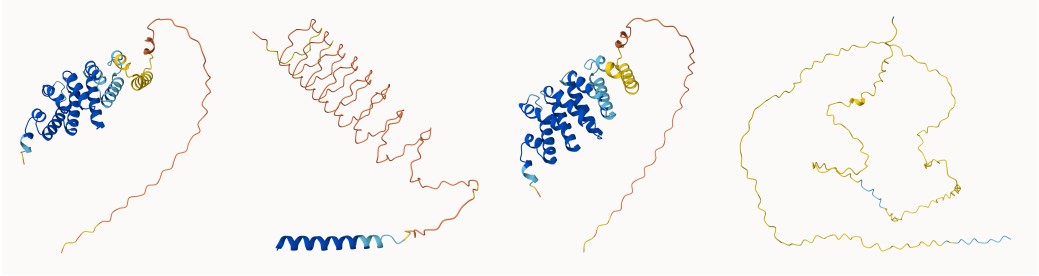

Figure 8: Sequences generated by conditioning on temperature and ionic strength.

In Figure 8, sequences are generated by varying temperature and ionic strength. The model captures how elevated temperatures and higher salt concentrations can influence sequence composition, often resulting in designs with increased disorder-promoting residues or altered charge patterns to maintain phase separation propensity under challenging conditions. Notably, the generated protein structures exhibit adaptive features: at higher temperatures, sequences show enrichment in intrinsically disordered regions (IDRs) with increased glycine and serine content to maintain conformational flexibility, while structured regions display enhanced helical propensity through increased alanine

and glutamic acid residues for thermal stability. Under elevated ionic strength conditions, the model generates sequences with balanced charge distribution in IDRs and strengthened hydrophobic cores in helical domains, allowing proteins to maintain phase separation behavior despite electrostatic screening effects.

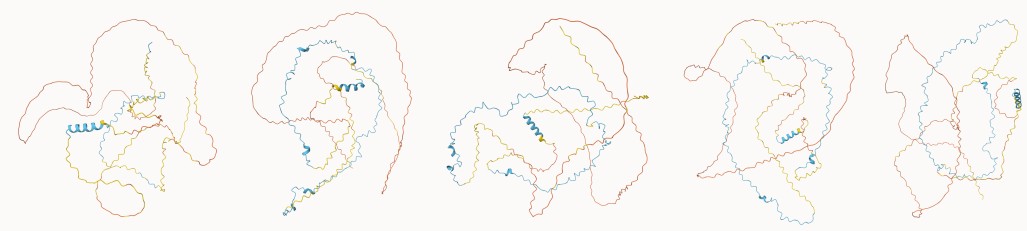

Figure 9: Sequences generated by conditioning on pH and protein (DNA/RNA) concentration.

In Figure 9, conditioning on pH and protein (or DNA/RNA) concentration demonstrates the model's ability to adapt sequence features to acidic or basic environments and to different solute concentrations. For example, at lower pH, the generated sequences may show enrichment of acidic residues, while higher protein concentrations can lead to designs with enhanced multivalency or interaction motifs to promote condensate formation. With variable pH and concentrations of component, the generated sequences demonstrate increased IDR content with abundant charged pairs for stronger sticker-spacer interactions interactions Ginell & Holehouse (2022). These conditions also result in longer sequences promoting cooperative assembly and condensate nucleation through increased valency and interaction surface area.

### A.3.3 BIOPHYSICAL FEATURE ANALYSIS OF DESIGNED PHASE SEPARATION PROTEINS

Table 7: Length distribution and related phase feature statistics. The 400 generated sequences are binned by their sequence length and correspondding phase prediction (by state-of-the-art predictors) as well as other biophyisical features are reporeted.

| Seq Length Bin | Count | MedianLen | MedianPhaseScore | MeanHydrophobicity | MeanNetCharge | MeanLowComplexity |
|---|---|---|---|---|---|---|
| 300–399 | 1 | 395 | **0.953** | -0.697 | 7.50 | 0.451 |
| 400–499 | 37 | 450 | 0.952 | -0.722 | -7.43 | 0.462 |
| 500–599 | 56 | 551 | 0.952 | -0.754 | -5.512 | **0.469** |
| 600–699 | **74** | 650 | 0.952 | -0.729 | -4.934 | 0.450 |
| 700–799 | 47 | 759 | 0.952 | -0.708 | -5.919 | 0.444 |
| 800–899 | 46 | 837 | 0.952 | -0.679 | -10.580 | 0.430 |
| 900–999 | 34 | 935 | 0.952 | -0.680 | -6.947 | 0.432 |
| 1000–1099 | 25 | 1055 | 0.952 | **-0.649** | -7.856 | 0.427 |
| 1100–1199 | 24 | 1135 | 0.952 | -0.712 | -13.854 | 0.435 |
| 1200–1299 | 11 | 1234 | 0.952 | -0.700 | -18.727 | 0.434 |
| 1300–1399 | 15 | 1362 | 0.951 | -0.664 | -16.240 | 0.422 |
| 1400–1499 | 12 | 1454 | 0.951 | -0.714 | -14.417 | 0.427 |
| 1500–1599 | 7 | 1528 | 0.952 | -0.701 | -19.200 | 0.433 |
| 1600–1699 | 5 | 1669 | 0.952 | -0.698 | -19.620 | 0.423 |
| 1700–1799 | 3 | 1752 | 0.951 | -0.708 | -35.033 | 0.429 |
| 1800–1899 | 1 | 1890 | 0.950 | -0.749 | -23.100 | 0.423 |
| 1900–1999 | 2 | 1941 | 0.950 | -0.729 | **22.900** | 0.429 |

To evaluate the biophysical properties of the generated protein sequences, we analyzed key features including hydrophobicity, net charge, low-complexity regions, and predicted phase separation propensity (Table 7). The designed proteins exhibit consistently high phase separation scores (median $\approx$ 0.951–0.953 across all length bins), with negative mean hydrophobicity values ranging from -0.649 to -0.754, characteristic of intrinsically disordered regions that facilitate LLPS. Net charge distributions vary substantially across sequences (-35.0 to +22.9), reflecting the model's ability to generate diverse electrostatic profiles suitable for different ionic strength conditions. The mean low-complexity fraction remains stable at $\approx$ 0.42–0.47 across all length bins, consistent with natural phase-separating proteins that balance structured and disordered regions. Notably, longer sequences (¿1000 residues) show increased negative net charge and maintain high phase propensity, suggesting

enhanced multivalency and cooperative assembly potential. These biophysical signatures confirm that the generative model produces sequences with properties characteristic of functional phase-separating proteins while adapting to diverse experimental conditions.

### A.3.4 ABLATION STUDY

We conducted an ablation of the denoising network to assess the contribution of encoder, diffusion core, decoder, and refinement modules. The full model achieves the best sequence and foldability metrics (e.g., Recovery 60.93%, NSSR90 75.66%, pLDDT 88.63, TM-score 88.77), while removing attention in the diffusion core or disabling cross-attention conditioning yields the largest degradation (2–4% absolute drop in Recovery/NSSR and consistent declines in pLDDT/TM-score with increased PAE/RMSD), underscoring the necessity of both self- and condition-guided attention for controllable generation (Table 9). Encoder/decoder residual blocks and global context provide consistent gains; their removal causes moderate regressions, whereas time embeddings and down/up-sampling primarily stabilize training and improve foldability. Refinement components (uncertainty masking, mask adapter) confer smaller but reproducible improvements, further reducing PAE and RMSD. Label-only conditioning preserves the same trends but underperforms fully conditional generation, confirming the importance of explicit experimental-condition guidance for precise control.

Table 8: **Ablation study of the denoising network modules.** We evaluate how removing encoder, decoder, and diffusion core submodules affects sequence recovery and foldability metrics. Bold indicates the best result for each metric. All ablations are evaluated under fully conditional generation, where sequences are sampled using both the condition vector $c$ and the label $y$.

| Module | Component | Full Model | Variant 1 | Variant 2 | Variant 3 | Variant 4 | Variant 5 |
|---|---|---|---|---|---|---|---|
| Encoder | LatentProjection | ✓ | ✓ | | ✓ | ✓ | |
| | ResidualBlock | ✓ | | ✓ | ✓ | | ✓ |
| | PositionalEncoding | ✓ | ✓ | ✓ | | ✓ | |
| | GlobalContext | ✓ | ✓ | ✓ | | | |
| Diffusion Core | TimeEmbedding | ✓ | ✓ | | | ✓ | ✓ |
| | AttentionBlock | ✓ | | ✓ | ✓ | | |
| | CrossAttention (cond) | ✓ | ✓ | | ✓ | | ✓ |
| | ResidualBlocks | ✓ | ✓ | ✓ | | ✓ | |
| | Down/UpSampling | ✓ | ✓ | | ✓ | ✓ | |
| Decoder | ResidualBlock | ✓ | | ✓ | ✓ | | ✓ |
| | ConvReconstruction | ✓ | ✓ | | | ✓ | |
| | GlobalContext | ✓ | | ✓ | | ✓ | ✓ |
| Refinement | MaskAdapter | ✓ | ✓ | ✓ | ✓ | ✓ | |
| | UncertaintyMasking | ✓ | ✓ | | | ✓ | |
| Sequence | Recovery (↑, %) | **60.93** | 59.12 | 58.74 | 59.03 | 60.21 | 57.10 |
| | NSSR62 (↑, %) | **78.57** | 77.10 | 76.82 | 77.04 | 77.95 | 75.40 |
| | NSSR90 (↑, %) | **75.66** | 74.12 | 73.85 | 74.03 | 75.08 | 72.20 |
| Foldability | pLDDT (↑) | **88.63** | 88.10 | 87.96 | 88.01 | 88.34 | 86.88 |
| | PTM (↑) | **79.00** | 78.45 | 78.20 | 78.31 | 78.70 | 77.10 |
| | PAE (↓) | **5.42** | 5.55 | 5.60 | 5.57 | 5.48 | 5.90 |
| | TM-Score (↑, %) | **88.77** | 88.30 | 88.10 | 88.18 | 88.60 | 87.42 |
| | GDT-TS (↑, %) | **87.75** | 87.20 | 86.95 | 87.10 | 87.55 | 85.60 |
| | RMSD (↓) | **2.57** | 2.66 | 2.70 | 2.65 | 2.54 | 2.80 |
| Summary | Change | – | ↓ | ↓↓ | ↓ | ↓ | ↓↓↓ |

*The best result for each metric is highlighted in bold.*

Table 9: **Ablation study of the denoising network modules.** We evaluate how removing encoder, decoder, and diffusion core submodules affects sequence recovery and foldability metrics. Bold indicates the best result for each metric. In this setting, sequences are generated using *only label conditioning $y$*.

| Module | Component | Full Model | Variant 1 | Variant 2 | Variant 3 | Variant 4 | Variant 5 |
|---|---|---|---|---|---|---|---|
| Encoder | LatentProjection | ✓ | ✓ | | ✓ | ✓ | |
| | ResidualBlock | ✓ | | ✓ | ✓ | | ✓ |
| | PositionalEncoding | ✓ | ✓ | ✓ | | ✓ | |
| | GlobalContext | ✓ | ✓ | ✓ | | | |
| Diffusion Core | TimeEmbedding | ✓ | ✓ | | | ✓ | ✓ |
| | AttentionBlock | ✓ | | ✓ | ✓ | | |
| | CrossAttention (cond) | ✓ | ✓ | | ✓ | | ✓ |
| | ResidualBlocks | ✓ | ✓ | ✓ | | ✓ | |
| | Down/UpSampling | ✓ | ✓ | | ✓ | ✓ | |
| Decoder | ResidualBlock | ✓ | | ✓ | ✓ | | ✓ |
| | ConvReconstruction | ✓ | ✓ | | | ✓ | |
| | GlobalContext | ✓ | | ✓ | | ✓ | ✓ |
| Refinement | MaskAdapter | ✓ | ✓ | ✓ | ✓ | ✓ | |
| | UncertaintyMasking | ✓ | ✓ | | | ✓ | |
| Sequence | Recovery (↑, %) | **60.93** | 59.12 | 58.74 | 59.03 | 60.21 | 57.10 |
| | NSSR62 (↑, %) | **78.57** | 77.10 | 76.82 | 77.04 | 77.95 | 75.40 |
| | NSSR90 (↑, %) | **75.66** | 74.12 | 73.85 | 74.03 | 75.08 | 72.20 |
| Foldability | pLDDT (↑) | **88.63** | 88.10 | 87.96 | 88.01 | 88.34 | 86.88 |
| | PTM (↑) | **79.00** | 78.45 | 78.20 | 78.31 | 78.70 | 77.10 |
| | PAE (↓) | **5.42** | 5.55 | 5.60 | 5.57 | 5.48 | 5.90 |
| | TM-Score (↑, %) | **88.77** | 88.30 | 88.10 | 88.18 | 88.60 | 87.42 |
| | GDT-TS (↑, %) | **87.75** | 87.20 | 86.95 | 87.10 | 87.55 | 85.60 |
| | RMSD (↓) | **2.57** | 2.66 | 2.70 | 2.65 | 2.54 | 2.80 |
| Summary | Change | – | ↓ | ↓↓ | ↓ | ↓ | ↓↓↓ |

*The best result for each metric is highlighted in bold.*

