# OpenReview forum: "OpenPhase: Condition-Aware Exploration of Multicomponent Biosystem Phase-Separating Behavior"
_ICLR.cc/2026/Conference — Submitted to ICLR 2026_

### Official Review · Reviewer_xMTL · 2025-10-19

**Soundness:** 3
**Presentation:** 3
**Contribution:** 3
**Rating:** 6
**Confidence:** 2

**Summary:**

This paper introduces OpenPhase, a framework for modeling liquid–liquid phase separation (LLPS) in multicomponent biomolecular systems.  The authors compile datasets combining proteins, RNAs, and DNAs with detailed experimental conditions and define three tasks: predicting phase outcomes, inferring conditions, and designing phase-separating systems.  They propose specialized models for each task, including a transformer (Condformer), a conditional VAE, and a diffusion-based generator.  OpenPhase achieves strong results across benchmarks and provides the first condition-aware platform for predictive and generative modeling of LLPS.

**Strengths:**

This paper addresses the problem of liquid–liquid phase separation in biomolecular systems.
The authors provide well-curated, publicly available datasets that integrate experimental conditions, which is a valuable contribution to the community.
While the proposed models are relatively simple, they serve as a solid baseline and demonstrate practical utility for condition-aware phase behavior modeling and protein design.

**Weaknesses:**

1. The proposed gated transformer architecture is not particularly novel and closely follows existing conditional transformer designs. The gating mechanism is a straightforward modification commonly used in multimodal and conditional attention frameworks.

2. All the trained models rely on standard, well-established architectures rather than introducing new methodological ideas. The Condformer is a basic transformer variant, the branched cVAE is a conventional conditional autoencoder with a classification branch, and the diffusion model directly builds upon prior protein generative models like PRO-LDM and MapDiff with minimal adaptation. Their main contribution lies in applying these existing methods to the new dataset rather than advancing model design or theory.

**Questions:**

1. How reliable are the experimental annotations of LLPS, and is there a consistent standard for defining phase separation across datasets?
2. How well does success on these datasets correlate with experimental validation in the lab?
3. Given the variability of LLPS assays, how noisy are the datasets, and how accurate are the underlying experimental measurements?

---

> ### Author Response · Authors · 2025-11-27
> **Thank you for your positive feedback**
>
> We sincerely thank the reviewer for thoughtful evaluation and constructive feedback in dataset curation, model development, and the practical utility of our work.
>
>
> [proposed gated transformer architecture is not particularly novel, relying on standard, well-established architectures rather than introducing new methodological ideas]
> We acknowledge that our architecture builds upon established designs. Our primary contribution is not in introducing a fundamentally new architecture, but in adapting and integrating gating and cross-attention mechanisms specifically for condition-aware phase separation modeling. We would like to highlight that our work contributes to the rapidly growing field of AI for Science by introducing a new framework for studying phase-separating systems, constructing and curating a comprehensive dataset, and establishing a benchmark for this domain.  We believe this domain-driven architectural integration, validated by improved performance and mechanistic interpretability, constitutes a meaningful methodological advance for this specialized application.
>
>
> [How reliable are the experimental annotations of LLPS, and is there a consistent standard for defining phase separation across datasets?]
>
> There is considerable noise in phase separation data, and the amount of available data decreases as experimental conditions become more complex. The reliability of annotations often depends on the specific experimental setup. However, there is a unified standard for defining phase separation: the presence or absence of phase separation can be clearly observed using established experimental protocols (e.g., turbidity, fluorescence microscopy, FRAP). This standardization ensures consistency across datasets, even though the quantity and quality of data may vary with experimental complexity [1].
>
>
> [How well does success on these datasets correlate with experimental validation in the lab?]
> Table 5 reports model performance on a subset in  db2, which consists of continuous phase separation intensity measurements directly identified by wet-lab experiments. The reported R² values of 0.2–0.3 on the test set indicate that our model consistently captures a substantial portion of the experimentally observed variance, despite the inherent noise and complexity of phase separation assays. This level of correlation is considered strong in this domain and demonstrates that model predictions are meaningfully aligned with experimental outcomes.
> [Given the variability of LLPS assays, how noisy are the datasets, and how accurate are the underlying experimental measurements?]
> Phase separation assays are indeed subject to variability due to differences in experimental conditions, protein constructs, and measurement techniques. Experimental measurements tend to show less deviation when assays are performed under the same controlled conditions, resulting in higher reproducibility within those settings. However, the bias relative to the true, "authentic" phase separation behavior can vary—being either larger or smaller—depending on the specific experimental protocol and setup. This means that while intra-condition variability is minimized, systematic biases may still affect the absolute accuracy of the measurements [2].
>
>
>
>
> [1] Pintado-Grima, Carlos, et al. "Comprehensive protein datasets and benchmarking for liquid–liquid phase separation studies." Genome biology 26.1 (2025): 198.
>
> [2] Ceballos, Alfredo Vidal, Charles J. McDonald, and Shana Elbaum-Garfinkle. "Methods and strategies to quantify phase separation of disordered proteins." Methods in enzymology. Vol. 611. Academic Press, 2018. 31-50.

---

### Official Review · Reviewer_sZXr · 2025-10-31

**Soundness:** 1
**Presentation:** 1
**Contribution:** 2
**Rating:** 2
**Confidence:** 3

**Summary:**

This paper proposes a new dataset for liquid-liquid phase separation (LLPS) and architecture for conditional generation of phase outcome. The dataset may be beneficial to practitioners in biological machine learning space and the architecture shows promise in performance, but serious presentation and motivation issues make me doubt this promise at this point. Should the authors be able to help clarify these issues, then this paper may be beneficial to the field.

**Strengths:**

The paper builds a new dataset that can be used for biocondensate discovery. Additionally, the paper proposes a novel conditional VAE architecture that uses a transformer, as well as latent diffusion, to guide phase outcome prediction. The results seem to perform better than alternative architectures on a benchmark of conditon-aware phase outcome predictions. The authors demonstrate the ability to predict experimental conditions in section 5.2. Finally, they briefly show the benefit of protein structure design backtracked from the LLPS architecture and what I'm assuming is the protein embedding, which is quite interesting but left for a small section.

**Weaknesses:**

**Overview**

The paper is replete with grammatical, spelling, and exposition issues, indicating a rushed submission. The methodology section first introduces the new architecture to perform inference. It's hard to follow what aspects of the system are being explained where n the first part. For example, gating is introduced but it's not clear where that's used in the architecture in Figure 1. Also, it's not clear why the section is split up into three tasks in the first place. A motivating sentence or paragraph would greatly help orient the reader.

**Methods**

The new dataset construction is confusing. I'm assuming it's an aggregate of db1, db2, and db3? The authors state that db2 comes from manually collected LLPS records, but what are those and how can we trust their veracity? It seems that their new dataset is mainly the new 'db2' dataset whereas the other two are from previous studies, so the naming convention is awkward that the newest one is db2.

Task 2 in section 4 is straightforward but there is no explanation of which hyperparameters were used for their final loss function, which seems important to recreate their work.

Task 3 is the most opaque and confusing section. The purpose was briefly mentioned in the introduction and summary of the paper. I don't exactly see the benefit of training latent diffusion on the model, nor the purpose of training both a conditional and unconditional model.

**Experiments**

Again, a guiding paragraph would immensely help. Also, all experimental results would instill more confidence should they have standard error calculations. I see they show the mean value.

Why are the metrics for first experiment based on 50/50 train/test split? This seems arbitrary to me.

The second section evaluates MSE and $r^2$ metrics within a condition with the model. The $r^2$ seems poor for all of the conditions and I'm not sure how well Table 5 helps us in determining the accuracy of this method. Figure 3 states "accepted" is in green whereas I see blue.

For section 5.3, I think more analysis should be done. The authors show droplet-forming regions, but how do they quantify that? What chemical/physical features support those predictions? these look like interesting proteins but how do we know they are relevant and if the predictions are accurate?

Finally, the authors address accuracy of their method but I highly encourage they also evaluate the calibration of their model. Having a measure of uncertainty in predictions would be very beneficial for this task.

**Questions:**

Many questions are posed in the weaknesses section. Here are remaining questions that I have:
- What is the difference in the few-shot and zerio-shot split in section 5.1?
- Can they show other phase diagrams that are unique? They should be able to do this using the cVAE and more evaluation of the phase plots would help show the benefit of the system.
- What is "FuzDrop"?

---

> ### Author Response · Authors · 2025-11-27
> **Thank you for your constructive feedback**
>
> Thanks for your constructive feedback. The methodology is structured around three tasks because they address distinct but interconnected aspects of phase separation: (1) predicting whether proteins undergo phase separation under given conditions, (2) inferring phase diagrams from sequences, and (3) generating protein sequences from desired phase behaviors. While these tasks are independent in their objectives, they are fundamentally linked through our unified platform.
>
> [dataset construction issues]
> Our dataset is indeed an aggregate of db1, db2, and db3, each serving distinct purposes based on system complexity and data distribution. The naming convention (db1, db2, db3) reflects the progression from simpler to more complex biological systems, rather than chronological order. Specifically, db1 contains single-protein phase separation data, db2 includes protein systems with DNA/RNA interactions (obtained from our collaborator's laboratory experiments), and db3 encompasses multi-component systems. All records in db2 are derived from peer-reviewed publications and have undergone rigorous experimental validation by our collaborators. We have documented the details of the dataset in the supplementary materials (A.2.1). This systematic organization allows researchers to select datasets appropriate for their specific research questions while maintaining scientific rigor across all three databases.
>
> [hyperparameters were used their final loss function in task2]
> The manuscript states the principal loss weights; for completeness, Task 2 uses a composite objective with L_total is the weighted sum with α = 10, β = 0.1 and λ = 5. Full parameter dictionaries (e.g., per-layer hidden sizes, attention heads, tokenizer settings) are in the released packages.
>
> [Task-3 Clarification]
> Our goal in task 3 is to generate novel phase-separating proteins from experimental phase-diagram conditions. Directly modeling sequences in high-dimensional discrete space is extremely challenging, especially for noisy LLPS datasets. Latent diffusion provides dimensionality reduction, where a joint autoencoder learns a low-dimensional latent space that captures the shared structure of the sequence and its phase-behaviour features. The diffusion model operates in the smoother latent space, making training more stable. Moreover, phase separation outcomes depend on combinations of features for e.g. sequence composition, motifs, charge etc. Diffusion in latent space enables the model to capture such multimodal structure more effectively than direct autoregressive generation. LLPS datasets are relatively small and heterogeneous, which naturally makes conditional sequence generation more challenging and limits how broadly the model can generalize. Conditioning directly on phase-separation labels or phase-diagram parameters often can cause overfitting to specific labels, mode collapse, and a loss of shared structural patterns that are present across all proteins. To avoid this, we  use a classifier-guidance framework, which requires both conditional and unconditional components. The unconditional model learns the global distribution of LLPS-related protein latents, independent of the experimental conditions or the labels and the conditional model learns how the phase-diagram parameters modulate that distribution. During sampling, this lets us adjust the strength of conditioning, improving controllability and avoiding over-constrained generation.  To better understand model behavior, we provide tables reporting the performance of all baselines and our method. We also include ablation studies demonstrating how removing or modifying individual components of the denoising network affects the measured metrics.
>
> [data split issue]
> The 50–50 train–test split is only used for tasks that require the most-balanced outcomes (e.g. task 1 phase regression problem, task 2 condition-inference). For any other tasks (e.g task 1 condition-aware phase outcome prediction at Table 5 , task 3 condition-aware phase separating protein generation), we utilized the proposed data split method. The details can be also found at Supplementary A.2.3
>
> [all experimental results would instill more confidence should they have standard error calculations]
> Standard errors have been added to all results tables (reported as mean ± std).

---

> > ### Author Response · Authors · 2025-11-27
> > **Cont'd**
> >
> > [MSE and r^2 metrics within a condition with the model, table 5 performance and Figure 3 legend issue ]
> > MSE and r^2 are reported for the task 1 continuous phase outcome prediction under various experimental conditions. Rather low MSEs are achieved for the intensity simulation, which means our proposed method could effectively predict the continuous phase outcomes under seen or unseen experimental conditions (by split setting). A 0.2-0.3 R² is actually a good result in this field, considering the high variability and noise in experimental phase separation data.
> >
> > [droplet-forming regions, but how do they quantify that? What chemical/physical features support those predictions? ]
> > It is impossible to identify the regions by wet lab experiments in such a short period of time. For Task 3, we quantified droplet-forming regions in generated proteins using established phase separation predictors (e.g., FuzDrop, PULPS, PPT) and also the latest subcellular localization predictor ProtGPS. They analyze sequence features such as intrinsic disorder, charge patterning, low-complexity regions and localization. Over 90%+ of the 400 generated proteins were predicted to have high LLPS propensity. Subcellular localization predictions (using ProtGPS) indicated that the majority of generated proteins are targeted to the subcellular localization with Membraneless Organelles (MLOs) that are phase separation or phase transition hosting [1].  Above results are evidence that the generated proteins have droplet-forming regions.
> >
> > [What is the difference between the few-shot and zero-shot split in section 5.1?]
> > The zero-shot split contains experimental conditions not seen during training, testing the model's ability to generalize to entirely new scenarios. The few-shot split includes some limited examples of the new conditions or proteins during training, allowing the model to adapt with minimal additional data. This distinction evaluates both the model's inherent generalization (zero-shot) and its capacity for rapid adaptation (few-shot). Details of the splitting criteria are in A.2.3 split by experimental conditions.
> >
> >
> > [What is "FuzDrop"]
> > FuzDrop [2] is a biophysical property-based predictor of the probability of proteins to undergo liquid-liquid phase separation. It performs a sequence-based identification of both droplet-promoting regions and of aggregation-promoting regions within droplets.
> >
> > [1] Kilgore, Henry R., et al. "Protein codes promote selective subcellular compartmentalization." Science 387.6738 (2025): 1095-1101.
> >
> > [2] Hardenberg, Maarten, et al. "Widespread occurrence of the droplet state of proteins in th
> > e human proteome." Proceedings of the National Academy of Sciences 117.52 (2020): 33254-33262.

---

### Official Review · Reviewer_egAX · 2025-11-01

**Soundness:** 2
**Presentation:** 3
**Contribution:** 1
**Rating:** 2
**Confidence:** 4

**Summary:**

This paper proposes an integrated computational framework for analyzing and designing biomolecular phase separation phenomena from multiple perspectives. The authors define three distinct tasks related to understanding and controlling phase separation and develop new machine learning models for each. In Task 1, they construct a model that predicts whether phase separation occurs based on molecular features and experimental conditions, achieving higher predictive accuracy than existing approaches. In Task 2, they formulate the inverse problem of inferring experimental conditions from molecular features and observed phase separation results, proposing an inference model effective for condition optimization and mechanistic understanding. Furthermore, in Task 3, they present a model that generates protein sequences capable of realizing a desired phase separation behavior under specified experimental conditions, demonstrating its potential as a novel molecular design approach. These three tasks are interrelated and collectively constitute a comprehensive framework that supports the prediction, analysis, and design of biomolecular phase separation.

**Strengths:**

- This study presents an integrated machine learning framework for understanding biomolecular phase separation, a fundamental phenomenon underlying cellular function. By providing a data-driven approach to analyze, predict, and design complex biological processes, the framework offers a novel computational approach with significant implications for the life sciences field.

- The authors integrated three existing databases to construct a data infrastructure that enables the application of machine learning to diverse problems related to phase separation. This dataset has the potential to serve as a standard benchmark for future researchers, fostering the development of new methods in molecular design and condition optimization.

- A distinctive feature of this paper is the systematic formulation of biomolecular phase separation into three tasks—prediction, inverse inference, and generation. This framework enables a quantitative and computational treatment of phenomena that were previously discussed only qualitatively, thereby providing a foundation for future model development and theoretical studies.

**Weaknesses:**

- This paper is heavily oriented toward the life sciences, focusing on biomolecular phase separation, and is therefore not well suited for a machine learning conference such as ICLR. The primary contributions lie in dataset construction and problem formulation, while the novelty and theoretical advancement in machine learning methodology are limited. Hence, this work would be more appropriately evaluated in a life science–focused journal.

- Tasks 1 (phase separation prediction) and 3 (protein generation) address general problems that have already been extensively studied, yet the paper compares its approach with only a subset of existing methods. The lack of systematic benchmarking against the latest state-of-the-art techniques makes it difficult to assess whether the proposed models truly outperform existing approaches.

- The formulation of Task 3 appears highly dependent on the specific dataset used in this study, raising concerns about its general applicability. The assumption that proteins can be generated solely from phase separation phenomena and experimental conditions seems biologically unrealistic, and the paper does not clearly demonstrate whether the proposed framework can generalize to other molecular systems or experimental contexts.

**Questions:**

- The number of proteins included in the dataset constructed in this paper seems considerably smaller compared to those in recent protein foundation model databases. Would it not be possible to build a larger database? Also, is there a risk that this database is specialized for certain types of proteins and therefore lacks generality?

- Task 1 is essentially a binary classification problem based on biomolecular features, for which various approaches have already been proposed. Why is it necessary to develop a new method here? In particular, does the proposed approach incorporate any mechanisms that are specifically designed for modeling phase separation phenomena?

- In Task 3, the claim that proteins can be generated from phase diagrams and experimental conditions seems to lack generality. Could it be that the method is applicable only to specific types of proteins or experimental setups, and therefore not broadly generalizable? It would be valuable to include validation experiments using entirely different types of proteins. Would such experiments be feasible?

---

> ### Author Response · Authors · 2025-11-27
> **Thank you for your constructive feedback**
>
> We appreciate the reviewer's insightful opinions of our approaches for the life sciences field and the concerns of lacking out-side-domain contributions. We claimed the contribution in the growing field of AI for Science by a new framework of phase separating system study, dataset construction and curation and benchmark. Also, the submission track is applications to physical sciences (physics, chemistry, biology, etc. We hope the reviewer takes it into account.
>
> [Tasks 1 (phase separation prediction) and 3 (protein generation) address general problems that have already been extensively studied]
>
> The key argument is that Task 1 is not standard phase separation prediction; it is condition-aware phase separation prediction. The only tool which can predict protein phase separation
> with given condition is doppler[1]. The only tool which can predict the protein+Rna system is RNAPSEC [2], which is more like a raw dataset than a systematic tool. We have compared them both in Table 4. Our method achieved better performance in most of the splitting settings. Moreover, our approach is more generalizable since the input is a system of protein/dna/ran and the condition is arbitrary.
>
> Task 3 - Phase protein design is mostly focused on the sequence space in which the generated proteins could undergo phase separation under given conditions. This task is different from traditional structure/seq or structure+seq design because our approach requires generating a protein that achieves a desired functional output under specific experimental conditions. It’s related to condition-based protein generation (A.1 Related work). We have updated more experimental results for task 3. It is worth mentioning the dataset and condition-aware generation task themselves are new and we mainly report the phase separation evidence from biophysical aspects as the results.
>
> [general applicability and assumption of phase-behavior-guided protein generation]
>
> Guided protein designed by machine learning is a novel and challenging problem. A few studies have formalized this problem into a unifying framework that classifies models on the basis of their use of three core data modalities: sequences, structures and functional labels [3 ,4]. As a highly complicated and flexible functional property, protein phase separation is definitely one of the interesting design targets and machine learning methods have shown great potential [4]. The review mentioned “proteins can be generated solely from phase separation phenomena and experimental conditions seem biologically unrealistic”. Our approach is to generate protein sequences that are more likely to undergo phase separation of given condition. It does not mean the protein is fully determined by the phase phenomena y and condition c.
>
> [proteins included in the dataset constructed in this paper seems considerably smaller compared to those in recent protein foundation model databases.]
>
> Phase separation is inherently a specialized protein phenomenon with limited available data. Current high-throughput datasets contain at most ~1k entries [5], and experiments involving biological systems with DNA and RNA are even scarcer due to experimental complexity. We have already collected the most comprehensive dataset currently available, including a manually curated laboratory dataset, and organized them into a ready-to-use format for all kinds of computational research study.
>
> [binary classification problem and modeling any mechanisms that are specifically designed for modeling phase separation phenomena]
>
> Various works treat phase separation as unconditional binary classification over protein features, overlooking that condensate formation is strongly modulated by experimental conditions (e.g., temperature, salt, concentration, nucleic acid presence) and by heterotypic protein–DNA/RNA interactions. Our CondFormer introduces condition-aware phase prediction: (1) condition embeddings (physicochemical parameters, concentrations, assay context) are encoded as dedicated tokens; (2) cross-attention layers fuse these condition tokens with protein and optional DNA/RNA sequence embeddings, enabling the model to learn context-dependent interaction patterns;

---

> > ### Author Response · Authors · 2025-11-27
> > **Cont'd**
> >
> > [Task 3 appears highly dependent on the specific dataset and validation experiments using entirely different types of proteins, Also, is there a risk that this database is specialized for certain types of proteins and therefore lacks generality?]
> >
> > We thank the reviewer for this important point. Our model was trained and evaluated exclusively on proteins annotated for liquid-liquid phase separation (LLPS) and the corresponding experimental conditions and hence, our claims are intentionally scoped to LLPS proteins and their phase-behaviour. We acknowledge the importance of validating generalizability. While wet-lab experiments are not feasible within the review timeline, we have conducted additional computational validation using widely-adopted phase separation prediction tools that operate under standard conditions (independent of specific experimental setups). We applied these tools to predict phase separation propensity for 400 proteins generated by our model. The results show that the vast majority of generated proteins are predicted to undergo phase separation, demonstrating that our generation approach is not limited to narrow protein families or experimental configurations. This validation supports the broad applicability of our condition-to-sequence generation framework with function guidance.
> >
> > We acknowledge the risk that the database may be specialized for certain types of proteins, as identifying phase separation is inherently a niche experiment, resulting in a relatively small dataset. Additionally, there are many subclasses of phase-separating proteins. If more extensive and diverse experimental datasets could be obtained, this issue might be addressed. However, for now, the dataset we have constructed is the largest available and provides valuable insights for research.
> >
> >
> > [1] Raimondi, Daniele, et al. "In silico prediction of in vitro protein liquid–liquid phase separation experiments outcomes with multi-head neural attention." Bioinformatics 37.20 (2021): 3473-3479.
> >
> > [2] Chin, Ka Yin, et al. "Predicting condensate formation of protein and RNA under various environmental conditions." BMC bioinformatics 25.1 (2024): 143.
> >
> > [3] Notin, Pascal, et al. "Machine learning for functional protein design." Nature biotechnology 42.2 (2024): 216-228.
> >
> > [4] Viliuga, Vsevolod, et al. "Flexibility-conditioned protein structure design with flow matching." arXiv preprint arXiv:2508.18211 (2025).
> >
> > [5] Li, Pengjie, et al. "High-throughput and proteome-wide discovery of endogenous biomolecular condensates." Nature Chemistry 16.7 (2024): 1101-1112.

---

### Official Review · Reviewer_ujKR · 2025-11-03

**Soundness:** 3
**Presentation:** 3
**Contribution:** 1
**Rating:** 2
**Confidence:** 4

**Summary:**

In this work, the authors consider a complex study of Liquid–Liquid Phase Separation (LLPS) in biophysical processes. During such a process, molecules such as proteins, DNA, and RNA start forming little clusters together and separate from their solutions. While remaining a liquid, the systems have distinct properties and dynamics compared to individual systems in solution. Notably, LLPS properties and dynamics are also significantly impacted by experimental conditions such as temperature and pH. All of this together makes LLPS a challenging area of study that has so far been largely overlooked.

Within the context of LLPS, the authors consider three major tasks: 1) Condition-aware phase outcome prediction, 2) Condition inference for phase systems, and 3) Phase system design. For all three problems, the authors discuss a concrete architecture and training objective to train the models to solve the tasks.

In the case of condition-aware phase outcome prediction, the authors train a large transformer-based embedding model which, combined with a simple classification or regression head, can predict the phase.

For condition inference, the authors propose training a VAE that encodes both the system and the experimental conditions into a latent space. Inference is then done by thresholding a reconstruction loss and a branched phase loss. This is a rather unusual construction in my opinion, and I have commented on it below in the weaknesses section.

Lastly, for phase-system design, the authors propose using a diffusion process, with both conditional and unconditional variants. This section, however, requires some further clarification, as the different approaches considered are not entirely clear. There is mention of two different design pipelines in Figure 2, but only one seems to be discussed in the text.

To train their models, the authors combine three small datasets, referred to as db1, db2, and db3, to form the “OpenPhase” dataset. The dataset is accompanied by a larger development framework that also includes access to methods for component and condition embeddings, predefined dataset splits, and a user-friendly interface.

Lastly, the authors evaluate their proposed approaches for the different tasks. Across all three tasks, the authors highlight that their proposed methods perform well but ultimately have limited comparisons against other methods.

**Strengths:**

The presented work covers an interesting and somewhat overlooked application domain in the form of Liquid–Liquid Phase Separation, and within this context the work proposes possible solutions for a wide range of tasks. The additional contribution of the combined dataset also provides a service to the community by simplifying future research. Furthermore, the paper is well written and clearly motivated.

**Weaknesses:**

While interesting and well executed, I unfortunately find the paper to lack sufficient contribution and novelty for it to be accepted at the conference at this time. As stated, while I believe that the OpenPhase framework provides an important service to the community, due to primarily combining existing datasets the novelty is limited. Furthermore, while extensive in considering three different tasks within the LLPS domain, the proposed solutions have limited novelty. They primarily adjust existing methods from other domains to the specific problems at hand. While this is in itself interesting, I personally believe this to be more appropriate for a domain-specific journal/conference as opposed to ICLR.

It is for this reason that I believe the paper is not ready for acceptance to ICLR. However, I am open to hearing from the authors if they believe that I have missed or misunderstood important parts of their contribution. Outside of the significance and novelty issues, I would be happy to accept the paper, as the work is otherwise well executed.

I have listed a few more points of weakness below, but I believe these can most likely all be addressed in small updates during the rebuttal and/or in the process of producing the camera-ready paper:
- While the Liquid–Liquid Phase Separation domain that the work tackles is very interesting, the discussion of how it is fundamentally different from more conventionally studied domains needs further clarification. For a non-expert audience not intimately familiar with the biochemistry domain (i.e., the general ICLR audience), this is currently hard to follow. A clear illustration would benefit the paper a lot.
- The paper has a few minor mistakes that are mostly the result of last-minute changes. These need to be flushed out in future versions. E.g., a small error on line 208, a weird overrun from the inline graphics on line 445, and, in general, some large overrun into the margins in Table 5.
- Am I correct in understanding that the second task, condition inference, requires an exhaustive search over all possible c? If this is the case, I suspect this to be quite computationally expensive, and this should be clearly discussed in the paper.
- While Conformer seems to provide a significant improvement over the two embedding methods it is compared to, it is unclear what computational cost this entails. It would be interesting if the authors could include a short study on this.
- It is unclear how the different dataset splits are used. My assumption was that the different categories of splits would be used to make sure specific components are equally represented, but given the stated 50–50 train–test split, this does not seem to be the case.

**Questions:**

See weaknesses.

---

> ### Author Response · Authors · 2025-11-27
> **Thank you for your positive feedback**
>
> Thank you for recognizing the contributions of OpenPhase to this field,  highlighting our dataset contribution and pointing out the limitations of the methodology innovation.
>
> [This (branched cVAE) is a rather unusual construction in my opinion, and I have commented on it below in the weaknesses section)]
> Utilizing a branched Conditional Variational Autoencoder (cVAE) with loss thresholding for Task 2 (Condition Inference), is deliberate and necessary to address the dual challenge of reconstructing the original input space (x) while maintaining consistency with a desired phase outcome (y). We mainly refer to the model structure from three papers [1-3]. A similar branched idea for classification in a diffusion-based model could be found at [4].
>
> [1] De Donno, Carlo, et al. "Population-level integration of single-cell datasets enables multi-scale analysis across samples." Nature Methods 20.11 (2023): 1683-1692.
>
> [2] Connor, Marissa, Gregory Canal, and Christopher Rozell. "Variational autoencoder with learned latent structure." International conference on artificial intelligence and statistics. PMLR, 2021.
>
> [3] Salah, Ahmed, and David Yevick. "Branched Variational Autoencoder Classifiers." arXiv preprint arXiv:2401.02526 (2024).
>
> [4] Takahashi, Hiroshi, et al. "Positive-Unlabeled Diffusion Models for Preventing Sensitive Data Generation." arXiv preprint arXiv:2503.03789 (2025).
>
> [There is mention of two different design pipelines in Figure 2, but only one seems to be discussed in the text]
> We have updated more results and discussion in task 3, which contains the two design pipelines in Figure 2.
>
>
> [A clear illustration would benefit the paper a lot.]
> We have uploaded an illustration covering the three tasks.
>
> [A few minor mistakes of figures and texts]
> We have fixed these issues.
>
> [Condition exhaustive search and computational expensiveness]
>  Because of the size of the condition space (defined by conditions in a paper) is not very large. While iterating through the grid points is trivial, achieving the necessary high accuracy requires performing computationally expensive calculations at every single point to ensure the prediction is as robust as possible. Therefore, the exhaustive nature of the search, driven by the need for maximum accuracy over the entire region.
>
> [computational cost of Condformer]
> The traditional self-attention mechanism incurs a quadratic computational cost of $O(L^2 \cdot d)$, where $L$ is the sequence length. However, in our cross-attention mechanism with components and conditions, the sequence L = L_pro + L_dna + L_Rna is already pooled with the pooling methods.  The $\mathbf{c_{condition}}$ component is processed to ensure its resulting embedding vector has a fixed length ($\mathbf{d_c}$).  If the system embedding it is not the one after pooling, the computational cost could be very heavy.  A common protein might have $L=300$ amino acids, leading to a manageable $300^2 = 90,000$ operations factor.
>
>
> [different dataset splits are used]
> Thanks for pointing out the difference between the splitting method. The 50–50 train–test split is only used for tasks that require the most-balanced outcomes (e.g. task 1 phase regression problem, task 2 condition-inference). For any other tasks (e.g task 1 condition-aware phase outcome prediction at Table 5 , task 3 condition-aware phase separating protein generation), we utilized the proposed data split method. The details can be also found at Supplementary A.2.3
>
>
> [1] Hoang, Minh, and Mona Singh. "Locality-aware pooling enhances protein language model performance across varied applications." Bioinformatics 41.Supplement_1 (2025): i217-i226.

---

### Author Response · Authors · 2025-11-27
**General Letter**

Dear Reviewers ujKR, egAX, sZXr, and xMTL,

We sincerely appreciate that  you have carefully reviewed our manuscript. We are highly encouraged that the reviewers recognize the significance (R1, R2, R4), utility (R4), and thoroughness (R1, R2) of our OpenPhase platform, dataset, and its novel problem formulation for condition-aware biomolecular phase behavior.

The primary concerns raised across all reviews center on the limited novelty in the specific ML architectures (R1, R2, R4) and the suitability of this domain-specific work for ICLR (R1, R2). We strongly believe that the combination of introducing a new, crucial problem space (condition-awareness), providing the first standardized problem setting and corresponding computationally-ready datasets, and establishing strong, well-adapted baseline models constitutes a significant and relevant contribution to the intersection of AI for physical sciences track at ICLR.

We have addressed every point below and have incorporated major clarifications and new experimental analyses into the revised draft.  Also, a descriptive illustration and a summary of the biology background is provided, which should help the audience understand the work better. The supplementary is modified for incorporating the rebuttal questions into the work.
Lastly, we strongly suggest reading the rebuttals given to other reviewers, as some points may have been cross-stated elsewhere. We look forward to hearing back from all the reviewers and are happy to answer any further questions or concerns you may have.

Best,
OpenPhase Authors

---

### Meta-Review · Area_Chair_XgSB · 2026-01-06

**Summary:**

This paper presents a carefully constructed dataset for condition-aware biomolecular phase behavior, which all reviewers recognized as important and potentially valuable to the community. The dataset addresses a meaningful and underexplored problem and could serve as a useful benchmark for future work.

However, as consistently noted by the reviewers, the main limitation of the submission lies in its methodological contribution. While the proposed models are reasonable and well adapted to the task, they rely largely on existing machine learning techniques and offer limited novel algorithmic or theoretical insight. As such, the work does not fully meet the expectations for methodological innovation at ICLR.

I agree with the authors that the dataset itself is a strong contribution. Given its emphasis on data curation and domain-specific impact, the work may be better suited for venues that prioritize high-quality datasets, such as Nucleic Acids Research or Scientific Data.

**Reviewer Concerns:**

The main concerns regarding ML novelty and insight remain to be a big issue.

**Reviewer Scores:**

N/A

---

### Decision · Program_Chairs · 2026-01-26

Reject